# Single cohesin molecules generate force by two distinct mechanisms

Georgii Pobegalov [1,2], Lee-Ya Chu[1], Jan-Michael Peters [3] & Maxim I. Molodtsov [1,2,3] ✉

Spatial organization of DNA is facilitated by cohesin protein complexes that move on DNA and extrude DNA loops. How cohesin works mechanistically as a molecular machine is poorly understood. Here, we measure mechanical forces generated by conformational changes in single cohesin molecules. We show that bending of SMC coiled coils is driven by random thermal fluctuations leading to a ~32 nm head-hinge displacement that resists forces up to 1 pN; ATPase head engagement occurs in a single step of ~10 nm and is driven by an ATP dependent head-head movement, resisting forces up to 15 pN. Our molecular dynamic simulations show that the energy of head engagement can be stored in a mechanically strained conformation of NIPBL and released during disengagement. These findings reveal how single cohesin molecules generate force by two distinct mechanisms. We present a model, which proposes how this ability may power different aspects of cohesin-DNA interaction.

Cohesin was originally identified for its role in physically connecting sister chromatids in proliferating cells. This sister chromatid cohesion is essential to ensure accurate chromosome segregation in mitosis[1–3]. In addition to sister chromatid cohesion, cohesin also organizes higher-order DNA structures in interphase cells by folding DNA into loops and topologically associating domains (TADs)[4–6]. This function has been implicated in gene regulation, recombination and other genomic processes that require accurate spatial organization of DNA[7–9].

The core of the complex consists of two ~50 nm long SMC1 and SMC3 coiled-coil subunits that dimerize at the globular hinge domain and transiently interact via ATPase head domains that form ABC-type ATP binding and hydrolysis sites (Fig. 1a)[10,11]. ATPase heads are also connected by a partially unstructured SCC1 kleisin subunit. The N- and C-terminal parts of SCC1 interact with SMC3 and SMC1 subunits, respectively. A fourth subunit, either STAG1 or STAG2, binds SCC1 and provides additional binding interfaces for interactions both within the cohesin as well as with other proteins and DNA. One such key interaction is with NIPBL-MAU2 (SCC2 in yeast and referred herein as NIPBL^Scc2), which also interacts with cohesin's ATPase heads as well as SCC1 and is required for both topological loading onto DNA and for DNA loop extrusion[12–17].

In vitro single-molecule studies indicated that cohesin can both topologically load onto DNA by entrapping it inside its ring structure as well as actively extrude DNA loops[15,16,18–22]. Both cohesin loading onto DNA and DNA loop extrusion are stimulated by ATP hydrolysis and are dependent on the presence of NIPBL^Scc2. However, loop extrusion does not seem to require topological loading as it occurs even when all three cohesin ring interfaces are covalently closed[16].

ATP binding to the cohesin SMC1 and SMC3 head domains promotes their engagement and forms two composite ATPase active sites[23,24], resulting in two ATP molecules being hydrolyzed during one cohesin cycle. High resolution cryo-EM and AFM studies revealed that cohesin adopts different conformations when associated with different nucleotide states[24–26]. Cryo-EM showed that in the ATP-bound state the two ATPase heads of cohesin are engaged and SMC3 and SMC1 coiled-coils are bent at the elbow allowing the hinge to interact with the SMC3 head[24–27]. In contrast, dynamic single-molecule FRET experiments indicate that after ATPase head engagement the coiled-coils do not stay bent, but may straighten causing the hinge to move away from the heads[24], which is different from the conformation seen in cryo-EM. In the absence of ATP, the two heads of cohesin disengage

[1]The Francis Crick Institute, London NW1 1AT, UK. [2]Department of Physics and Astronomy, University College London, London WC1E 6BT, UK. [3]Research Institute of Molecular Pathology (IMP), Vienna BioCenter, Vienna 1030, Austria. ✉e-mail: m.molodtsov@ucl.ac.uk

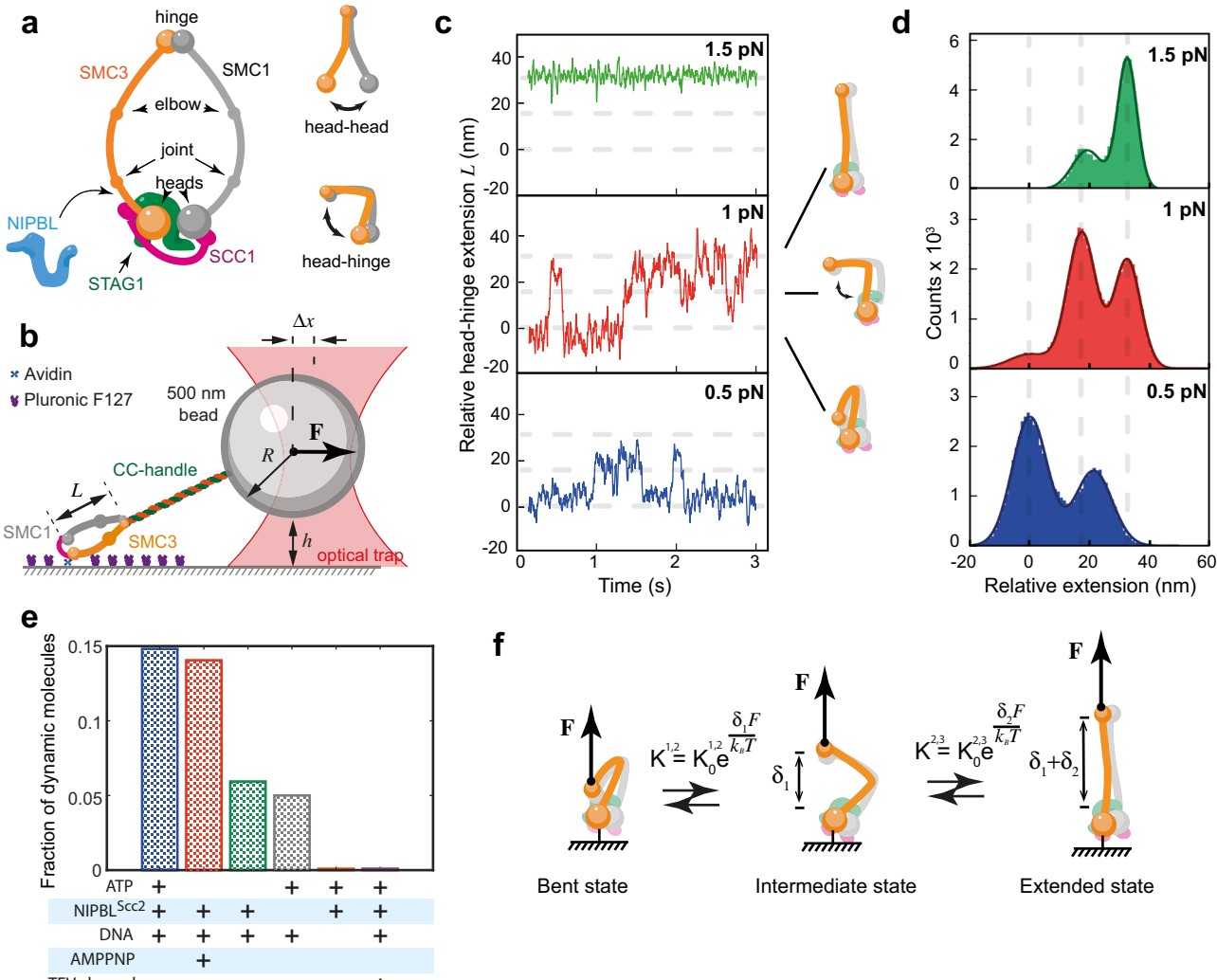

**Fig. 1 | Cohesin head-hinge movement is driven by Brownian fluctuations.**
**a** Schematic representation of a human cohesin complex and a cartoon showing its conformational changes. **b** Schematic of the experimental optical tweezers assay. The head-hinge distance is denoted ($L$). CC-handle is a passive coiled coil ~100 nm long. The bead displacement from the center of the optical trap ($\Delta x$), bead radius ($R$) and the height of the bead above the coverslip ($h$). **c** Representative traces of the relative head-hinge distance ($L$) measured at 0.5, 1 and 1.5 pN in the presence of ATP, DNA and NIPBL$^{Scc2}$. Cartoons of cohesin conformations corresponding to specific states are shown for the 1 pN trace. Gray lines are visual guides separated by 17 and 15 nm, respectively. **d** Cumulative histograms of the relative head-hinge distance at 0.5, 1 and 1.5 pN in the presence of ATP, DNA and NIPBL$^{Scc2}$ from all

experiments that passed alignment selection criteria ("Methods"). Source data for (**c**) and (**d**) are provided as a Source Data file. **e** Fraction of dynamic cohesin complexes is shown for different experimental conditions. The total number of head-hinge complexes accepted for analysis for each conditions was (from left to right: $n_{ATP} = 195$, $n_{AMPPNP} = 42$, $n_{noATP} = 17$, $n_{noNIPBL}^{Scc2} = 20$, $n_{noDNA} = 23$, $n_{+TEV} = 24$). **f** Three state model of cohesin bending. Conformations of cohesin shown on the figure are inferred from the measured head-hinge distance. Based on panels (**c**) and (**d**), $\delta_1 = 17$ nm, $\delta_2 = 15$ nm. Position of SMC1 is unknown and shown as an illustration. The steady state distribution between the states depends on the external force and is determined by the equilibrium constants shown on the figure.

and move away from each other[24,25] forming a conformation that has also been observed in the related SMC complex condensin[28]. When the heads are disengaged, cohesin can alternate between the bent conformation in which the SMCs are bent and the hinge touches the ATPase heads and an open conformation when the hinge is detached from the heads and coiled-coils are only partially bent or extended[19,24,28]. Thus, there are two major conformational changes characteristic to both cohesin and condensin: head-hinge movement driven by SMC coiled-coil bending and head-head engagement and disengagement (Fig. 1a). However, how these rearrangements mechanistically facilitate DNA loop extrusion and topological loading of SMC complexes remains poorly understood.

In this work we used optical tweezers to characterize the energetics of conformational changes in human cohesin complexes. We

show that bending of the SMC arms occurs in two distinct steps of ~15 and 17 nm. The transition rates depend exponentially on the external force and the movement stalls at forces higher than 1 pN, consistent with it being driven by thermal Brownian fluctuations. We also show that SMC head engagement and disengagement proceed in ATP dependent single ~10 nm steps. The rates of transitions are independent of the external force and can overcome forces of up to 15 pN. Our molecular dynamic (MD) simulations suggest that force generated by the head-head movement can be stored as mechanical stress in a strained form of NIPBL$^{Scc2}$ and released after ATP hydrolysis. We propose that mechanical forces generated by these conformational changes have roles in the initiation and elongation phases of the loop extrusion process and we discuss implications of the energetics of cohesin's conformational changes for understanding the mechanics of cohesin as a molecular machine.

## Results

### Head-hinge movement is driven by diffusive motion between three states

To determine the energetics of the head-hinge conformational change in cohesin, we immobilized individual molecules of human cohesin on the surface of a microfluidic flow-cell via the SMC3 head (Fig. 1b). Next, we applied controlled forces to the hinge domain via a 500 nm size polystyrene bead attached to the hinge by a ~110 nm long passive coiled-coil protein linker using an optical trap (Supplementary Fig. 1a). In a typical force-clamp experiment, we applied constant force to the molecule and monitored the change in the head-hinge distance ($L$) at this force. Due to the relatively large bead size comparing to the overall length of the molecular complex, we determined $L$ from the geometrical relationship between the position of the bead, the bead radius, and its height above the coverslip (Fig. 1b, See Methods for details).

In the standard reaction buffer, in the presence of 1 mM ATP, 5 ng/µL Lambda DNA and 10 nM NIPBL[Scc2] and at 0.5 and 1 pN of force cohesin head-hinge distances showed transitions between multiple states. The higher force of 1 pN gave better signal to noise ratios, which allowed us to distinguish three clear states (Fig. 1c, d, Supplementary Fig. 1b). At this force out of 28 dynamic molecules, 15 spent enough time in all three states to be analysed quantitatively. In comparison, when NIPBL[Scc2] was left out (with ATP and DNA still present), most molecules were static (19 out of 20 molecules). As an additional control, we performed experiments in high salt buffer containing 750 mM NaCl. Such high salt concentration screens electrostatic interactions[29] and we expected the activity of cohesin in these conditions to be strongly reduced. Indeed, we found that none of the molecules tested showed any dynamic changes in 750 mM NaCl ($n$ = 37 molecules, see Methods).

Dynamic molecules in standard conditions (50 mM NaCl) showed changes in distances of up to ~32 nm, suggesting that this is the maximum amplitude with which the hinge can move away from the ATPase heads. This measurement is consistent with the distance that the hinge could travel based on our negative stain EM data (Supplementary Fig. 1c), estimated from the available structures (Supplementary Fig. 1d)[11,24–27,30], and extracted from our EM images (Supplementary Fig. 1e). Therefore, we conclude that extended state observed in our experiments corresponds to cohesin molecules that have conformations with extended SMCs, while the shortest conformation corresponds to molecules in which the coiled coils are bent.

Interestingly, transitions between bent and extended states frequently occurred via an intermediate third state. The measured distances between the hinge and the SMC3 head indicate that in this state SMC coiled coils are only partially bent with the hinge being approximately half-way between fully bent and fully extended states (Fig. 1c). Histograms of traces obtained at 1 pN revealed three distinct peaks as molecules spent more than half of their time in either the intermediate or the extended states (Fig. 1d). The distance between peaks revealed that the hinge traveled ~17 nm between the bent and intermediate states and additional ~15 nm from the intermediate to the fully extended state.

At a lower force of 0.5 pN occasional transitions to the fully extended state were too brief to be reliably detected and the histogram combined from all molecules analyzed at this force showed only two peaks separated by ~20 nm. We reasoned that the first peak corresponds to cohesin spending approximately 70% of the time in the fully bent state and the second peak is a sum of the intermediate and extended states which are poorly resolved at this force (Supplementary Fig. 1f, see Methods). Interestingly, this closely resembles a distribution of the head-hinge distances of the yeast condensin that toggles between folded and open states[31], possibly indicating of similar conformational dynamics for cohesin at 0.5 pN force. Overall, our results indicate that under low-force conditions cohesin spends little time in the fully extended state and exists most of the time in the fully

bent state. At 1 pN the balance shifted towards intermediate and extended states. At forces of 1.5 pN and above cohesin is fully extended and shows no detectable transitions, suggesting that bending of the hinge to the heads cannot overcome forces larger or equal to 1.5 pN.

Next, we studied the effect of NIPBL[Scc2], ATP and DNA on the head-hinge transitions between the bent and extended states. Cohesin was most active in the presence of all NIPBL[Scc2], ATP and DNA. We found that the number of cohesin molecules demonstrating transitions between bent and extended states was significantly reduced and at least 3-fold lower when one of the components ATP, NIPBL[Scc2] or DNA was left out during the measurement (Fig. 1e, Supplementary Fig. 2a). We also took advantage of the 3x TEV site artificially engineered into the SCC1 subunit of our cohesin construct to study the effect of the kleisin cleavage on the head-hinge dynamics. Cleavage of cohesin at this site by TEV protease abrogates DNA loop extrusion[16] and removes topologically loaded cohesin from DNA[22]. We verified that the latter does indeed take place in our experimental conditions (Supplementary Fig. 2b, c) and therefore proceeded to study the effect of the SCC1 cleavage in our experiments. After we cleaved SCC1 with TEV just prior to the measurement, we could no longer observe head-hinge dynamics even in the presence of all components NIPBL[Scc2], ATP and DNA (Fig. 1d, Supplementary Fig. 2a), suggesting that the intact kleisin is required for the head-hinge dynamics and presumably for the activity of the whole complex. Finally, we tested if we could detect head-hinge dynamics in the presence of a non-hydrolysable ATP analog AMPPNP. Surprisingly, we found that in the presence of AMPPNP as well as both DNA and NIPBL[Scc2], the head-hinge dynamics was almost as efficient as in the presence of ATP at 1 pN force (Fig. 1e, Supplementary Fig. 2a). Since AMPPNP cannot provide input of the chemical energy into the system, this indicates that in the ATP-bound state the head-hinge dynamics is driven by random thermal fluctuations rather than the energy provided by ATP.

To further test this idea, we sought to extract the rates of transitions between the bent, intermediate and fully extended states and study their dependence on the external force. This is because in a system where movement is driven by thermal fluctuations, the rates of transitions between states must depend exponentially on force via Arrhenius factor $\exp(F\delta/k_BT)$, where $F$ is the external force and $\delta$ is the length of the molecule length change. To extract the rates, we fitted the histograms shown in Fig. 1d to the three-state model with thermally driven transitions between them as shown in Fig. 1f. We found that the steady state distribution of times that cohesin spends in each state agrees with this model and does indeed depend exponentially on the external force. The distributions for both ATP and AMPPNP experiments were well fitted by the exponential factors $\exp(F\delta_1/k_BT)$ and $\exp(F\delta_2/k_BT)$, where $\delta_1$ = 17 nm for the bent to intermediate states transitions and $\delta_2$ = 15 nm for the intermediate to extended transitions are taken directly from the single-molecule measurements (Supplementary Fig. 2d, e). This further corroborates the idea that transitions between three head-hinge states are driven by Brownian fluctuations between these states.

Moreover, the fits to the three-state model allowed us to extrapolate our results to zero force and predict the dynamics of the head-hinge movement in the absence of the applied load. The fitting suggests that in the absence of force ~95% of the time cohesin molecules spend in their bent state and only ~5% in the intermediate and extended states. Thus, our experiments show that head-hinge movement is driven by thermal fluctuations and in the presence of ATP, SMC coiled-coils can unbend but spend most of the time in the bent state when no external force is applied.

### Head-head movement is powered by ATP and generates up to 15 pN force

Next, we measured energetics associated with the head-head interaction using the same approach as above (Fig. 2a). In the presence of ATP

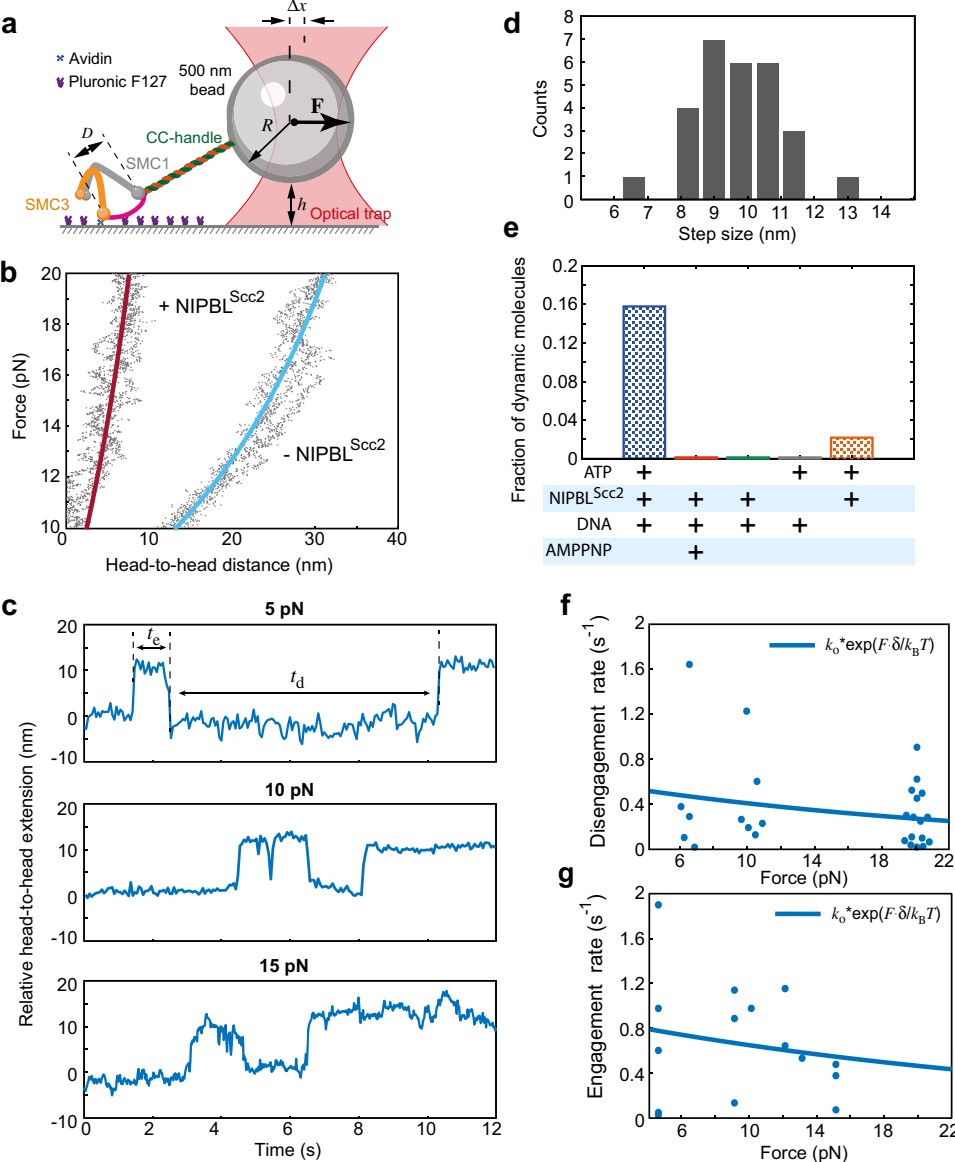

**Fig. 2 | Force generation by the cohesin head-head engagement. a** Schematic of the experimental optical tweezers assay. The SMC head-head distance is shown as *D*. Other notations follow Fig. 1b. **b** Relative head-head distances measured as function of external force in the absence and presence of NIPBL[Scc2]; dots, experimental values; solid lines are worm-like-chain fits with the following parameters: no NIPBL[Scc2]: $L_P = 2$ nm, $L_0 = 120$ nm; plus NIPBL[Scc2]: $L_P = 8$ nm, $L_0 = 23$ nm, where $L_P$ and $L_0$ are the persistence and contour length correspondingly. **c** Example of head-head movements measured by the optical *t*rap at 5,10 and 15 pN force. $t_e$ and $t_d$ on the 5 pN traces show times until engagement and disengagement events respectively that are used to calculate rates in (**g**) and (**f**). **d** Histogram of steps extracted from all

head-head distance measurements measured in the range 5–20 pN. **e** Fraction of dynamic cohesin complexes (i.e., ones that showed at least one engagement and one disengagement event) against external force. The total number of complexes that were accepted for analysis for each condition from left to right is ($n_{ATP} = 57$, $n_{AMPPNP} = 30$, $n_{noATP} = 24$, $n_{noNIPBLScc2} = 17$, $n_{noDNA} = 46$). **f** Cohesin head-head disengagement (opening) rate as a function of external assisting force. The solid line shows exponential fit with $k_O$ & $\delta$ as a free parameters. The values are discussed in the text. **g** Cohesin head-head engagement rate as a function of hindering force. Notations follow (**f**). Source data for this figure are provided as a Source Data file.

but without NIPBL[Scc2], cohesin molecules did not show any distinct movements between SMC heads, consistent with measurements of these movements by single-molecule FRET[24]. Rather, molecules responded to changes in the applied force like elastic elements stretching over 30 nm under ~20 pN of force (Fig. 2b).

When we added both ATP and NIPBL[Scc2], the stiffness of the head-head linkage strongly increased such that forces up to 20 pN did not lead to any significant elastic extension for the majority of the molecules tested (35 out of 43 molecules, Fig. 2b). This is consistent with cryo-EM and smFRET data showing that in the nucleotide bound state NIPBL[Scc2] interacts with both SMC heads and might therefore provide mechanical stiffness to the head-head linkage[24–27].

We noticed that in the presence of both NIPBL[Scc2] and ATP as force was increased in these experiments, some force-distance traces showed step-like transitions separated by ~10 nm distance (Supplementary Fig. 3a), which could be explained by the ATPase head engagement or disengagement events during the measurement. To gain more insight into how rates between these transitions depend on the external force, we performed experiments in which the change in the distance between the ATPase head domains was recorded under a constant force, similar to the approach we used for studying the head-hinge transitions. We detected abrupt transitions in the head-head distance separated by ~10 nm at forces in the range between 5 and 20 pN (Fig. 2c, d). These movements and their magnitude are

consistent with the movements between SMC1 and SMC3 head domains in the presence of ATP and NIPBL[Scc2] detected by smFRET[24]. The amplitude of the movement is also consistent with the post-hydrolysis state of cohesin[25] as well as with the apo-bridged state of condensin in which ATPase heads are separated[28]. Therefore, the two states observed in our experiments correspond presumably to cohesin conformations associated with engaged and disengaged SMC heads. Thus, our experiments revealed that the head engagement and disengagement movements can occur under mechanical forces of up to 15 pN, which is significantly stronger than 1.5 pN required to completely stall the head-hinge movement. We note that in these experiments external force stretches the molecule and therefore assists disengagement and opposes engagement (Fig. 2a).

Next, we tested the effect of NIPBL[Scc2], ATP and DNA on the head-head dynamics under external force and found that all three components were required for the ATPase heads to engage against external force. As shown in Fig. 2b, NIPBL[Scc2] greatly increases the stiffness of the complex in this configuration, suggesting that it may provide a mechanical link between the two heads and function as a mechanical scaffold that allows generation of the force when the heads engage. Consistent with this idea, we did not observe any head engagement events against the external force in the absence of NIPBL[Scc2] (Fig. 2e, Supplementary Fig. 3b). Without ATP, we detected occasional single disengagement events (4 out of 24) which were never followed by an engagement (Supplementary Fig. 3c). Thus, no engagement against external force was observed without ATP (Fig. 2e). Similarly, we found that the head-head engagement against force was much less efficient when DNA was absent (Fig. 2e). Finally, unlike in the head-hinge data, we could not detect dynamic head-head traces in the presence of the non-hydrolysable ATP analog AMPPNP (Fig. 2e, Supplementary Fig. 3b). Thus, we show that ATP and NIPBL[Scc2] are both required for the head-head dynamics under external force. This indicates that the energy stored in ATP may potentially be used to perform mechanical work required to move ATPase heads closer to one another against the hindering force applied by the optical trap.

To test if this was the case, we considered two different physical mechanisms that could in principle explain head-head engagement/disengagement under external force. The first mechanism is similar to the head-hinge movements: the engaged and disengaged ATPase heads may form a two-state system and transitions between the two states could be driven by random thermal fluctuations, albeit happen at forces higher than the head-hinge transitions. In this case, similarly to the head-hinge dynamics, the head-head transition rates between open and closed head-head states should depend exponentially on the external force: the rate of the head disengagement should increase with external force, while the rate of the head-head engagement should exponentially decrease. Alternatively, the head-head movement may be the result of a conformational change that is induced by ATP binding and is strong enough to engage the SMC ATPase heads against external force. This would be similar to a power stroke mechanism because in this case, external force should produce much less effect on the transition rate constants since energy change associated with transitions would depend on the chemical change associated with the nucleotide state rather than on the energy of thermal fluctuations[32].

To distinguish between these two possibilities, we measured the dependence of the head-head engagement and disengagement rates on the external force. We extracted the rates directly from the individual traces that showed transitions at different forces (see Methods). The ATPase head disengagement rates appeared to be independent of the external force (Fig. 2f) and direct fit to an exponential yielded the mechanical distance between the two states to be nonsensical −0.15 nm, thus ruling out exponential dependence of the disengagement rate on the force (Fig. 2f). Similarly, head-head engagement rates decreased only gradually with external force and never occurred

against forces above 15 pN (Fig. 2g). Direct exponential fit yielded the mechanical distance between the two states to be 0.1 nm, inconsistent with the 10 nm steps showed by individual traces (Fig. 2c, d). Thus, neither engagement, nor disengagement rate depend exponentially on the external force given the mechanical distance between the states of 10 nm. Therefore, we conclude that the head engagement/disengagement is not driven by the thermal fluctuations, but rather it could only be explained by a chemical change driving the transitions between engaged and disengaged states. Given the dependence of the transitions on ATP, these results suggest that ATP is likely the source of energy, which allows the ATPase head domains to engage against the external force and produce mechanical work.

## Energy of the head-head movement can be stored in the conformation of NIPBL[Scc2]

Our results are consistent with previous data that NIPBL[Scc2] is required for the SMC head-head movement[24]. Curiously, when the ATPase heads are engaged, cryo-EM data showed that NIPBL[Scc2] appears to be bent when compared to its X-ray structure[25–27]. It is therefore conceivable that the change of the NIPBL[Scc2] conformation is coupled to the movement of the ATPase heads. According to this hypothesis, NIPBL[Scc2] would adopt a bent conformation when the ATPase heads are engaged, and a more extended conformation seen by X-ray crystallography when the heads disengage (Fig. 3a). Consistent with this, a structural analog of NIPBL[Scc2] in a related complex condensin (Ycs4), also undergoes similar conformational changes depending on the ATP state of the SMC heads[33,34].

Since the conformation of NIPBL[Scc2] determined by x-ray crystallography is more relaxed, we hypothesized that the bent form observed in the ATP bound state may be less energetically favorable. This raises the possibility that NIPBL[Scc2] may act as a mechanical spring: it is pushed into the bent state by the engaging SMC heads, and it straightens and releases energy when the heads disengage. Since our experiments suggest that head engagement can generate mechanical force, we reasoned that the force required to bend NIPBL[Scc2] may be generated by head engagement, as schematically shown on (Fig. 3a).

To determine whether the bent form of NIPBL[Scc2] could store energy, we turned to MD simulations and calculated the free energy difference associated with the NIPBL[Scc2] conformational change when it transitions from its relaxed X-ray structure to the bent cryo-EM structure observed in cohesin when both heads are engaged.

In cohesin structures in the ATP-bound state, both human NIPBL[Scc2] (PDBID:6WG3) and its homolog SCC2 in yeast (PDBID:6YUF and PDBID:6ZZ6) have similar conformations and all appear bent with ATPase heads engaged[25–27]. However, neither yeast nor human relaxed structures are available. To obtain the relaxed conformation we used the structure of *Chaetomium thermophilum* SCC2 (PDBID:5T8V) and built a homology model using a *S. pombe* SCC2 sequence (called Mis4 in this species) and the same method as in ref. 25 (see Methods).

The bent conformation of NIPBL[Scc2] is seen in cryo-EM to make multiple salt bridges with the DNA phosphate backbone, while the relaxed structure is only available without DNA. Therefore, to disentangle the effect of the DNA on the energy difference that could potentially be stored in the bent conformation of NIPBL[Scc2] we sought to perform and compare MD energy calculations with and without the DNA. To perform the simulations without DNA, we simply removed DNA from the cryo-EM structure. For simulations with DNA, we started with the available cryo-EM structure that contained DNA and used it to dock DNA into the relaxed NIPBL[Scc2] structure to obtain the relaxed NIPBL[Scc2] with DNA.

Once the structures were obtained and equilibrated to find their stable states, we performed MD simulations and calculated the free energy change associated with the transition between the bent and relaxed structures based on the weighted histogram analysis method (WHAM). Note, that the exact trajectory along which the structure is

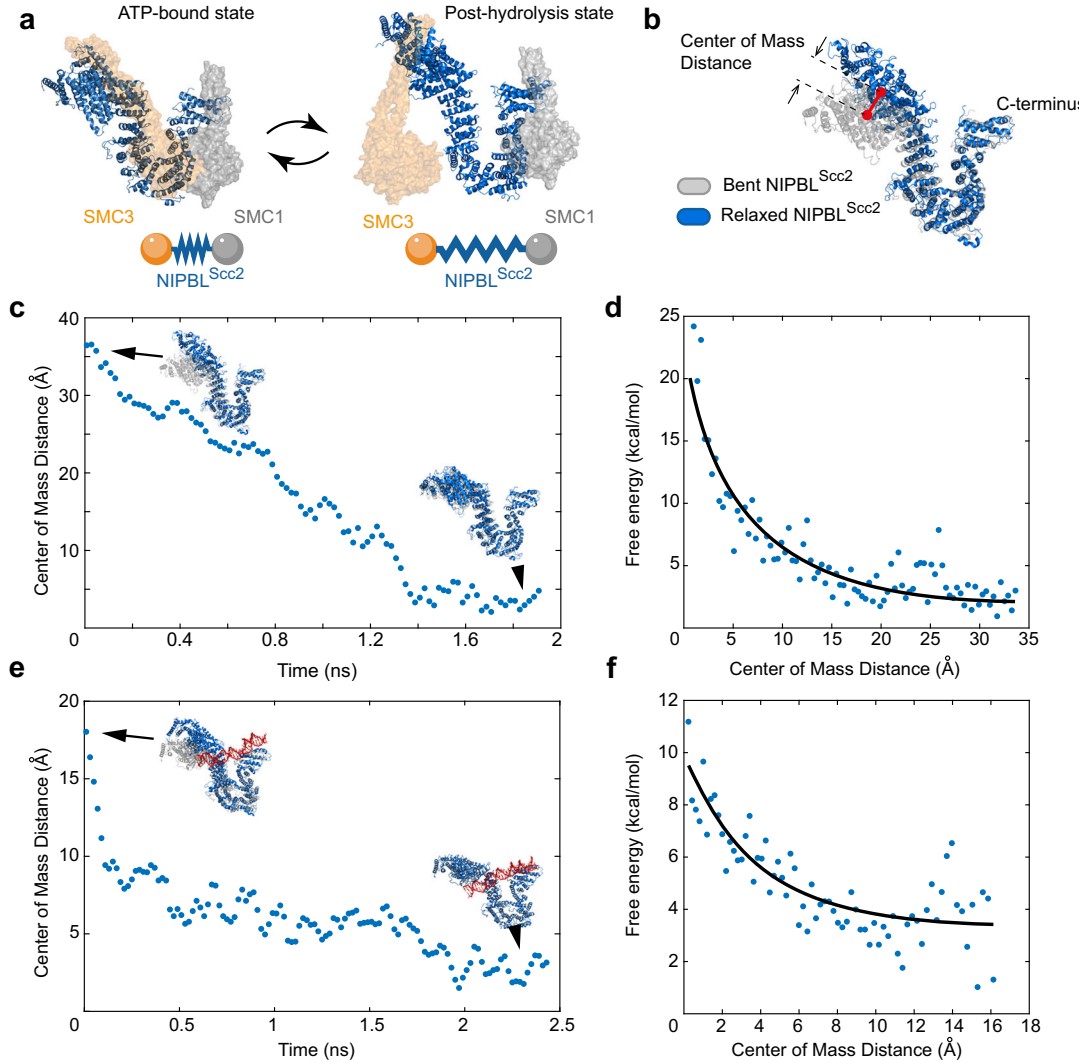

**Fig. 3 | NIPBL$^{Scc2}$ stores energy in its bent conformation. a** Cohesin and NIPBL$^{Scc2}$ in the state when SMC ATPase heads are engaged and their hypothetical conformation in the post hydrolysis state predicted based on the relaxed crystal structure form. Cartoon at the bottom represents cohesin loader as a mechanical spring, which is loaded when heads engage and releases energy when heads disengage. **b** Bent (gray) and relaxed (blue) NIPBL$^{Scc2}$ were aligned at C-terminal residues 587–1321. The center of mass distance between N terminal residues 1–586 is shown. **c** Change in center of mass distance as relaxed NIPBL$^{Scc2}$ is pulled to the bent conformation without DNA. Blue is the molecule, which is being pulled to the target conformation shown in gray. **d** Free energy change corresponding to data from (**c**) as a function of the center of mass distance. **e, f** Same as (**c, d**) but in the presence of DNA. Solid lines in (**d**) and (**f**) are trendlines and for illustration only. Source data for this figure are provided as a Source Data file.

changed from one state to another is not important because the difference in the free energy between the bent and relaxed states does not depend on the path used to arrive from one state to another. To monitor the progress of simulations and visualize the transition between relaxed and bent forms of NIPBL$^{Scc2}$, we aligned the C-termini of the molecule (residues 569–1321) and calculated the center of mass (COM) distance between their N termini (residues 1–568) (Fig. 3b). In the beginning of the simulation the COM distance between the relaxed and bent forms without DNA was ~37 Å and it decreased as the simulation progressed. After ~1.5 ns COM plateaued, and we stopped the simulation after another 0.5 ns to ensure that the final bent conformation of the molecule has been reached (Fig. 3c). The weighted histogram analysis revealed that the maximum energy required to bend NIPBL$^{Scc2}$ in the absence of DNA was ~18 kcal/mol (or 130 pN·nm, Fig. 3d).

With DNA the initial COM distance between the relaxed and bent forms was ~18 Å. This initial difference is smaller than without DNA because when we docked DNA to the relaxed NIPBL$^{Scc2}$ structure and performed the energy minimization after docking to equilibrate the

structure, the resulting relaxed structure with DNA became already slightly bent (Supplementary Fig. 4). After we performed MD to bend this structure further into the bent state in the presence of DNA, we found that the bent NIPBL$^{Scc2}$ was ~6 kcal/mol more energetic than the relaxed form (Fig. 3e, f). Thus, our MD simulations support the idea that the bent form of NIPBL$^{Scc2}$ has higher free energy than the relaxed form both with and without DNA, but the difference in the presence of DNA is smaller. We also find that the value of the free energy change required to bend NIPBL$^{Scc2}$ is comparable with the work that head-head movement can produce that we determined in our experiments, which had a value of 22 kcal/mol (10 nm × 15 pN = 150 pN·nm), and, therefore, head engagement could cause bending of NIPBL$^{Scc2}$ and increase its internal energy. Interestingly, our simulations show that significantly less energy is required to bend NIPBL$^{Scc2}$ when DNA is present. This may explain our findings that the head-head engagement against force is much more efficient in the presence of DNA (Fig. 2e).

The energy spent on the head-head engagement and NIPBL$^{Scc2}$ bending should come from binding of two ATP molecules to the heads of to SMC1 and SMC3. The exact amount of energy released from two

ATP molecules during one cohesin cycle is currently unknown. Typically, energy available from ATP hydrolysis is estimated to be between 7.3 and 15 kcal/mol (between 50 and 100 pN·nm per one ATP)[35,36] and it depends on the reaction conditions and the molecular mechanism of action. Similarly, energy can be released during the ATP binding step, which precedes hydrolysis. Although latter mechanism is different, the magnitude of the energy released is generally estimated to be in the same range[37,38]. Thus, binding of two ATP molecules to cohesin would be sufficient to generate the head-head force and to power the ATPase head engagement and NIPBL[Scc2] bending. Our simulations also suggest that DNA binding can contribute to NIPBL[Scc2] bending by lowering the free energy gap between the relaxed and bent conformations. Thus, ATP binding to the cohesin complex and the following formation of additional contacts between DNA and NIPBL[Scc2] presumably lead to the generation of the ATPase head engaged state with a much lower free energy. Our simulations suggest that the free energy made available from this reaction is partially converted to the energy of the bent NIPBL[Scc2] and partially remains available to perform additional mechanical work – in our case to move the bead out of the trap. Once ATP is hydrolyzed, heads disengage, and the cycle of engagement/disengagement may repeat.

## Discussion

Cohesin is a molecular machine that folds genomic DNA into loops and topologically associated domains by a mechanism, which is likely shared with those used by other SMC complexes[39]. Unraveling this mechanism is important for understanding how living organisms evolved ways to organize their DNA, and to control gene expression and recombination. However, mechanical details of how cohesin works as a molecular machine remain poorly understood. This is arguably because we lack mechanistic understanding of how chemical energy is transformed by cohesin to generate mechanical force required to move and rearrange DNA. In this study we applied mechanical force directly to single active molecules of cohesin and discovered that the head-head and hinge-head conformational changes in cohesin generate different amounts of mechanical force via distinct mechanisms.

Our experiments revealed the mechanism that drives the bending of the SMC coiled coils. Recent cryo-EM structures of ATP-bound human and yeast cohesin in the presence of NIPBL[Scc2] and DNA revealed a conserved conformation in which the SMC coiled coils are bent at the elbow region, allowing the hinge to interact both with NIPBL[Scc2] and the head-proximal part of SMC3[25–27]. The distance between the heads and the hinge in these structures is ~14 nm, while high-speed AFM and smFRET revealed that this distance can get smaller to within few nanometers for dynamically active cohesin[24]. The full extension of cohesin is achieved when the hinge and heads separate, resulting in their maximum distance of ~48 nm. These distances agree with the maximum change of the head-hinge distance that we measured in our experiments (Fig. 1d). Therefore, our assay detects dynamic bending of cohesin SMC coiled coils in their full range, from the bent to a fully extended state.

Unexpectedly, we discovered that transitions between the bent and extended conformations occur via an intermediate state corresponding to the movement of the hinge of ~17 nm, which is approximately halfway between fully bent and extended forms. We found that transitions between bent and extended states at 1 pN force require all: NIPBL[Scc2], ATP and DNA. This was unexpected because the head-hinge dynamics was previously observed without ATP or NIPBL[Scc2][24]. Our data show that this is because SMC coiled-coils are flexible (Supplementary Fig. 1f) and, therefore, at low forces they undergo random movement that leads to detectable change in the head-hinge distance. However, under externally applied force, we show that coordinated head-hinge movement does require all components. Interestingly, we found that transitions under external force also occur in the presence of AMPPNP,

which suggest that they are driven by Brownian fluctuations rather than energy of ATP. Consistently with this, we found that the transition rates between the bent, intermediate and fully extended states depend exponentially on the external force in both ATP and AMPPNP experiments, which suggests that the movement is best accounted for by a three-state model in which the transitions are driven by random Brownian fluctuations. The mechanism is also consistent with small absolute values of the force that inhibit the transitions.

The forces that lock the head-hinge distance in the fully extended state and prevent transitions are close to the forces that have been reported for stalling DNA loop extrusion and compaction by cohesin[15,18,40], condensin[20,41,42] and SMC5/6 complexes[43,44] compatible with the idea that the head-hinge movement is involved in translocation of DNA during loop extrusion. Consistent with this, blocking the head-hinge movement after loop extrusion was initiated stops loop extrusion[24]. The exact mechanism of how the head-hinge movement drives loop extrusion is, however, unknown, and a number of models have been proposed[19,24,31,45–49]. Our data show that the head-hinge movement does not generate strong forces and therefore favors models in which movement is driven by biased diffusion of the DNA, mediated by transient interactions of the hinge and the SMC ATPase heads.

Our experiments revealed, that as opposed to the head-hinge movement, ATPase head engagement, involves a single step of ~10 nm that can generate significant force of up to 15 pN. The size of the step is consistent with recent smFRET data[24], as well as the finding that it requires the presence of both ATP and NIPBL[Scc2]. We showed that AMPPNP does not support head-head dynamics under external force and that the rates of transitions between engaged and disengaged states do not depend on the external force, and therefore, are not consistent with the exponential dependence on the external force (Fig. 2f, g). This shows that the driver of the head engagement is not random Brownian movement, but presumably the chemical change associated with ATP binding, which is followed by the head engagement. In combination with our data that the engagement can work against significant mechanical load of up to 15 pN with almost no dependence of the rate on external force, these finding are compatible with the idea that head engagement generates a power stroke. The difference with other molecules is that power strokes in cytoskeleton motors and other molecular machines have been associated with directional movement, but whether or how head-head dynamics can enable unidirectional movement of cohesin remains to be understood. Interestingly, power stroke has been proposed to work during the ATPase head engagement in related to cohesin ABC transporter proteins (e.g., ref. 50). The force generation during engagement in both classes of molecules should require mechanical connection between the heads, and we speculate that in cohesin this role may depend on NIPBL[Scc2], which greatly increases the mechanical stiffness of the head-head linkage (Fig. 2b).

Force generation by engaging ATPase heads is also consistent with NIPBL[Scc2] changing its conformation from relaxed to bent when the heads engage (Fig. 3). Our interpretation is that after the ATP binding, the disengaged state of the cohesin/NIPBL[Scc2] system becomes energetically unfavorable. The subsequent conformational change coupled with NIPBL[Scc2] bending brings the heads together to a new state in which both heads are engaged and stabilized by the bent NIPBL[Scc2]. This increases the energy of NIPBL[Scc2], but decreases the overall energy of the cohesin/NIPBL[Scc2] complex. The difference between the new engaged state and the old state of the whole system provides the free energy that can be used to perform mechanical work, and, in our experiments, generated the force that pulled a microbead out of the laser trap. Thus, we propose that ATP binding provides free energy, which generates movement resulting in heads engagement, bends NIPBL[Scc2] and resists external mechanical force. Energy provided by ATP binding and not hydrolysis is not uncommon. Similarly, ATP

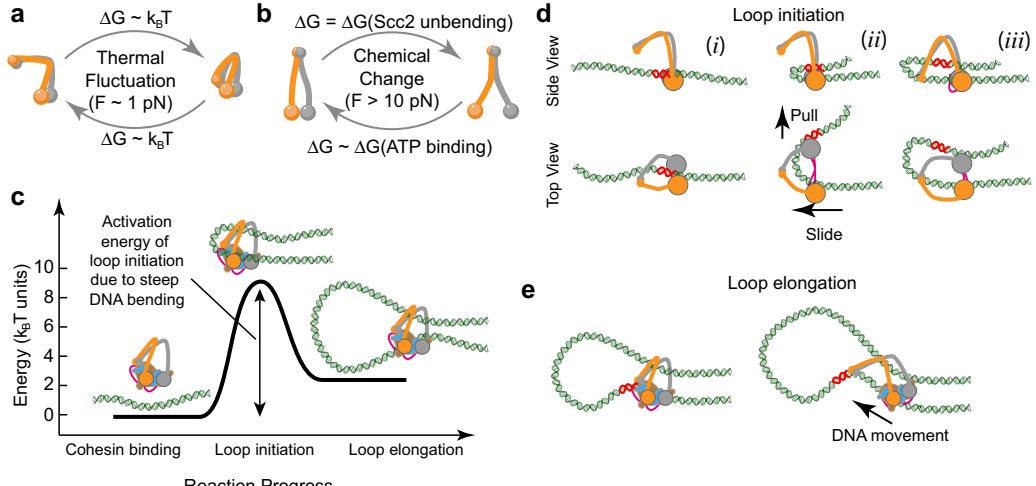

**Fig. 4 | Different force generating modes of cohesin are proposed to drive loop initiation and elongation. a** Free energy change associated with the head-hinge bending is on the order of $k_BT$ and it cannot generate significant force. **b** Head-head movement can generate strong forces suggesting that this conformational change is powered by the free energy of ATP binding. **c** Initial stages of loop extrusion require strong bending of DNA, which increases energy of the system and presents an activation barrier for the loop initiation. **d** A proposed mechanism for generating DNA protoloop by SMC head disengagement. (i) When SMC heads are engaged,

they both bind DNA. (ii) as heads disengage SMC1 head forms tight contact with DNA and pulls it away from SMC3 head. Because SMC3 allows sliding of DNA only in one direction, DNA forms a loop. (iii) Once the cycle is complete, another ATP binds causing SMC1 and SMC3 engagement. Cohesin is reset and an initial loop is created. **e** Once initialized, DNA loop can be elongated by movement between the head and the hinge. In (**d**) and (**e**) red segment marks discrete position on DNA to assist visualization of its movement.

binding provides energy for the conformational change in a related ABC transporter protein, while ATP hydrolysis has zero free energy change[37].

It has recently been proposed that a power stroke may be required to push DNA into the SMC ring[34,51], but the experimental evidence for this has been lacking. Our experiments show that head-hinge and head-head movements generate force via different mechanisms and suggest that ATP binding to cohesin may generate power stroke, which can overcome significant mechanical force. However, how these conformational changes and force generation in cohesin are coupled to the movement of cohesin on DNA remains unknown and require further investigation.

Our MD simulations revealed that the strong force generated by the head engagement could force the NIPBL^Scc2 to adopt bent conformation, observed in cryo-EM structures. Using MD simulations, we showed that the bent NIPBL^Scc2 is highly energetic and could serve as a loaded spring that can facilitate the separation of the two heads after ATP hydrolysis and ADP release. Although our optical tweezer experiments can only hinder head engagement and not disengagement, our data raise an interesting possibility that the energy released upon unbending of NIPBL^Scc2 may subsequently be utilized by cohesin to produce mechanical work as the two SMC heads disengage and move away from each other. Thus, NIPBL^Scc2 can potentially store the energy generated during the head engagement and release it during disengagement. Together our observations show that unlike thermally driven head-hinge movements, transitions between SMC head domains may work as a molecular machine and can generate significant forces due to free energy available from ATP. Thus, different energy sources drive head-hinge and head-head movements resulting in their distinct force generating properties as illustrated on Fig. 4a, b.

What may be implications of these two different force-generating mechanisms occurring in one molecule? Recent AFM imaging of condensin revealed molecules associated with very small loops of DNA, showing that DNA that is strongly bent, possibly representing states soon after loop extrusion has been initiated[31]. Although DNA molecules are very flexible at the chromosomal scale, this is not the case at the scale of single cohesin molecules. The persistence length of DNA is ~50 nm, and at this scale bending of DNA is associated with

increasing its elastic energy[52]. Therefore, loop initiation is a reaction that has an activation barrier, which at least partially consists of the energy required to bend DNA (Fig. 4c). Cohesin could potentially overcome this activation barrier spontaneously due to thermal fluctuations, but this is likely inefficient and slow: single-molecule measurements show that spontaneous bending of DNA at this scale occurs at the time scale between minutes and tens of minutes[52,53]. Therefore, one potential function of cohesin's power stroke may be to assist DNA bending at the initial stages of the loop extrusion.

Simulations using our recently developed molecular-mechanical model of cohesin[19] suggests that it is indeed possible for the ATPase head-head movement to generate enough force and drive loop initiation (See Methods and Supplementary Fig. 5). Assuming in the starting position that DNA is straight and positioned between the engaged SMC1 and SMC3, ATPase head disengagement can lead to two possible outcomes. If only one ATPase head remains bound to DNA, or both ATPase are bound but cannot generate force, no DNA bending would occur (Supplementary Fig. 5a). However, if both heads remain bound to DNA, and can generate force during disengagement, one ATPase head would pull DNA away from the other. If DNA can only slide perpendicular to the direction of the head-head movement this would inevitably lead to the formation of a DNA loop (Fig. 4d, Supplementary Fig. 5b). Thus, force generated by the SMC head-head movement could facilitate the initial DNA loop formation.

Once the loop has been formed, elongation could be driven either by the combination of the head-head and the head-hinge movements or by the head-hinge movement alone. For example, as the hinge bends towards ATPase heads, it can contact the initial DNA loop which just formed by the mechanism described above (Fig. 4e). Unbending of SMC coiled coils would then pull on DNA and lead to the loop elongation as was proposed in ref. 19. Alternatively, DNA could first be bound by the hinge, then be translocated to the ATPase heads by bending of the coiled coils[24], and only then be bent to initiate DNA looping by head. Thus, head-head and head-hinge conformational changes in cohesin could have distinct roles that can do both initiate and elongate DNA loops.

Given the importance of DNA organization, cohesin must have evolved controllable and reliable ways to initiate and elongate DNA

loops, and we propose how these two different activities can be accomplished by one cohesin molecule. While our model can explain both loop extrusion initiation and elongation, we cannot exclude that ATPase head dynamics that can generate significant amount of force may also be involved in other activities of cohesin related to loop elongation, DNA loading and unloading and translocation.

In conclusion, our results provide force measurements of mechanical transients in single cohesin molecules. We show that mechanical force is an important and distinct aspect of cohesin chemistry, and it has strong implications for understanding mechanistical details of cohesin as a molecular machine.

## Methods

### Microfluidic flow-cell

Microfluidic flow-cells were prepared as previously described[19] with minor changes. Flow cells were assembled using parafilm sandwiched between a silanized cover slip and a glass slide in which two holes were drilled and a metal tubing (New England Small Tube Corp) was glued using an epoxy glue (Devcon) to form an inlet and an outlet. Slides were cleaned by sonication in ethanol, 1 M NaOH and MilliQ water, dried at 100 °C on a heating plate and plasma cleaned for 5 min. Cover slips (Marienfeld, High-precision, 24 × 60 mm) were cleaned by sonication in acetone, ethanol, 0.1 M NaOH and MilliQ water (5 min for each step), dried at 100 °C on a heating plate and plasma cleaned for 10 min. After that the coverslip surface was silanized by incubating for 1 h in 5% dichlorodimethylsilane (Sigma-Aldrich) dissolved in heptane. Finally, cover slips were sonicated in chloroform twice for 5 min followed by 5 min sonication in MilliQ water and blow-dried using compressed air.

### Beads functionalization with avidin

200 μL of carboxyl coated polystyrene beads (500 nm diameter, Bangs Lab) were dissolved in 1 mL of Activation buffer (0.1 M MES pH 6.0, 0.5 M NaCl), sonicated for 5 min, and washed by microcentrifugation at 10,000 g for 7 min and supernatant removal. Beads were resuspended in 1 mL of Activation buffer, washed again, and resuspended in 1 mL of Activation buffer. 4 mg of EDC (Thermo Scientific) and 11 mg of sulpho-NHS (Thermo Scientific) were dissolved in 200 μL of Activation buffer and added to the beads followed by immediate mixing by vortexing for 20 s. The mixture was incubated at room temperature for 25 min while constantly rotating. The reaction was quenched by adding 1.5 μL of β-mercaptoethanol (40 mM, Sigma Aldrich). The beads were pelleted by centrifugation and resuspended in 1 mL of PBS. 10 mg of PEG(COOH) (Rapp Polymere GmbH) were dissolved in 100 μL of PBS, added to the beads and reacted for 2 h at room temperature while constantly rotating.

Subsequently the beads were washed three times by centrifugation and resuspension in 1 mL of PBS. The beads were resuspended in 1 mL of Activation buffer, sonicated for 5 min and washed in Activation buffer three times. Reaction with EDC and sulpho-NHS and quenching with β-mercaptoethanol were performed as at the previous stage. The beads were pelleted and resuspended in 1 mL of PBS followed by addition of 200 μL of Avidin-DN (Vector Laboratories). The reaction was carried out at room temperature while constantly rotating for three hours and quenched by washing 4 times in 1 mL of Tris-HCl 50 mM pH 7.5. The beads were stored at 4 °C.

### Cohesin surface immobilization

A flow-cell was connected to a syringe pump (Harvard Apparatus, Pico Plus Elite 11) and filled with PBS. 50 μL of biotinylated-BSA (Thermo Scientific, 0.05 mg/mL in PBS) were introduced and incubated for 15 min, followed by a 400 μL wash with PBS. 100 μL of Pluronic 1% solution in PBS were incubated for 20 min and washed with 400 μL of PBS. The flow-cell was further passivated by incubation of 50 μL of BSA (UltraPure, Thermo Scientific, 1 mg/mL in PBS) for 30 min followed by 400 μL PBS wash. Subsequently, the flow cell was incubated with 30 μL of Avidin DN (Vector Laboratories, diluted 1:10 in PBS) for 20 min, washed 2 times with 400 μL of PBS and equilibrated with buffer W500 (PBS, NaCl 500 mM, β-Casein 0.4 mg/mL (Sigma Aldrich), UltraPure BSA 0.4 mg/mL, DTT (Sigma Aldrich) 5 mM).

50 μL of human cohesin complexes with the head and the hinge domains labeled with biotin and HaloTag respectively, diluted to 1 nM in buffer W500 were incubated in the flow-cell for 10 min and washed 3 times with 200 μL of buffer W500. To block non-reacted Avidin the flow-cell was incubated with 100 μL of Biotin (Sigma Aldrich, 0.25 mM in buffer W500) for 10 min and washed with 200 μL of buffer W200 (PBS, NaCl 200 mM, β-Casein 0.4 mg/mL, UltraPure BSA 0.4 mg/mL, DTT 5 mM).

500 nm Avidin coated beads were coupled to the Myosin coiled-coil handle (CC-handle) by mixing the beads diluted 10 times with the CC-handle (10 nM final concentration) in a 100 μL reaction containing buffer W200. The mixture was sonicated for 1 min and incubated at room temperature for 5 min. Biotin was added to a final concentration of 0.25 mM, the beads were sonicated for 1 min and incubated for 5 min once again. After that the beads were washed 3 times by microcentrifugation at 10,000 g for 3 min, supernatant removal, and resuspension in 100 μL of buffer W200. Finally, the beads were resuspended in 35 μL of W200, sonicated for 1 min and introduced into the flow-cell at 10 μL/min. After 20 min of incubation, the unbound beads were removed by washing the flow-cell twice with 200 μL of W200 at 100 μL/min. The flow-cell was further supplied with 50 μL of 1 mM ATP (Sigma Aldrich), 5 ng/μL Lambda DNA (New England Biolabs) and 10 nM human NIPBL in buffer R (Tris-HCl pH 7.5 50 mM, NaCl 50 mM, MgCl₂ 1 mM, DTT 5 mM, BSA 0.25 mg/mL, 0.2 mg/ml glucose oxidase, 35 μg/ml catalase and 4.5 mg/ml dextrose) and measurements were started immediately.

For experiments with SCC1 cleavage, the flow cell was supplied with 50 μL of TEV protease (New England Biolabs) diluted 15X in TEV buffer (Tris-HCl 40 mM, DTT 1 mM) after washing the unbound beads with W200 buffer. The reaction was incubated for 10 min prior to addition of 50 μL of 1 mM ATP, 5 ng/μL Lambda DNA (New England Biolabs) and 10 nM human NIPBL in buffer R, after which measurements were started immediately (see Supplementary Methods for details).

### Single-molecule force-spectroscopy

Cohesin tethers were first analysed using tethered particle motion microscopy. For this, images of individual beads were recorded at 100 Hz for 10 s and the bead position in each frame was determined by custom made ImageJ-based script that tracks the COM of the bead image. 2D distribution of the bead position was analysed and only the beads showing a symmetry value (ratio between the minor and major amplitudes) and root mean square greater than 0.8 and 80 nm respectively were selected for further analysis (Supplementary Fig. 6a, b).

For the head-hinge measurements a cohesin-tethered bead was held in a single optical trap (JPK, Nanotracker) with the typical trap stiffness of 0.017 pN/nm. The Z position was adjusted so that the bead was placed just 60 nm above the surface of the flow-cell. To find this distance the position of the surface was first determined by monitoring the trapped bead while moving the coverslip up using the piezo stage. As soon as the coverslip touches the bead, it pushes it out from the trap along Z and the image of the bead becomes highly sensitive to the axial position of the piezo-stage. Supplementary Fig. 6c shows that we could determine the axial position of the coverslip using this method with ~10 nm precision using video tracking of the bead. Once the position of the coverslip was determined, we moved the coverslip down to 60 nm to set the desired distance between the bead and the coverslip.

The force measurement was started by first stretching cohesin in one direction with 1.5 pN force and then switching to stretching along the opposite direction with forces varying from 0.5 to 1.5 pN. Stretching was performed by moving a high-precision piezo stage while holding the bead in a stationary optical trap. The tether anchor point was determined as a midpoint between the stage position at −1.5 pN and 1.5 pN of force. Coordinates of the piezo stage in a force clamp mode at stretching forces of 0.5, 1 and 1.5 pN were recorded at 2 kHz and further analysed using JPK Data analysis software, OriginPro 2021 and MATLAB. The head-head measurements were done in a similar way, and the trap stiffness was chosen to be in the range 0.1–0.2 pN/nm and kept constant in each experiment.

When tested at forces above 20 pN, in more than half of the cases (10 out of 18 molecules tested with high forces) the head-head distance showed a second transition from 10 nm distance to ~30 nm (Supplementary Fig. 6d). Since this is the maximum distance between SMC heads in the absence of NIPBL$^{Scc2}$, we reasoned that the second transition at forces above 20 pN may be caused by a complete or partial dissociation of NIPBL $^{Scc2}$. Therefore, we focused our analysis on the 10 nm engagement and disengagement head movements that occur at lower forces. To produce the histogram of steps (Fig. 2d) individual traces were fitted with a step-type or stair-type function[54]. Transition rates between 10 nm steps extracted from experiments that showed the second step were included in the data on Fig. 2d when the applied force was 20 pN or below.

To obtain rates in Fig. 2f, g we used the same fitting algorithm and extracted times between individual engagement and disengagement events which are shown on Fig. 2c, 5 pN trace as an example. Each dot on Fig. 2f represents an inverse of such time (s⁻¹) for the trace with the corresponding force. Thus, disengagement rates correspond to the inverse times between engagement and engagement events (i.e., time passed after one engagement event until one disengagement event). Engagement rates are the times between the disengagement and engagement events respectively.

## Data analysis

Raw traces recorded at 2 kHz in a force-clamp mode were smoothed down to 20 Hz using adjacent averaging and corrected for the bead displacement from the trap center. Coordinates of the piezo stage were recalculated into the actual length of the cohesin tether as follows (Supplementary Fig. 6e):

$$L_c = \sqrt{x^2 + (r + h)^2} - (r + cc) \qquad (1)$$

where

$L_c$ – length of cohesin tether;
$x$ – piezo stage displacement;
$r$ – bead radius: 250 nm;
$h$ – height of the bead above the surface: 60 nm;
$cc$ – length of the CC-handle: 110 nm.

Although both, the bead diameter and the axial distance between the bead and the coverslip contributed to the uncertainty in determining the tether length, the relative change in the tether length remained insensitive to less than 10% for variations in the bead height and diameter possible in our experiments (Supplementary Fig. 6f).

The error in determining the relative change in the cohesin length was only limited by the resolution of our instrument, which was higher for higher stiffnesses and typically better than 4 nm for experiments at 1 pN force, and better than 2.5 nm for experiments at forces of 5 pN and above (Supplementary Fig. 1b).

We also estimated the error with which we could potentially measure the absolute length of cohesin, which could be useful for determining the state of the static tethers. This is more challenging as the measurement of the absolute length directly depends on both bead diameter and bead axial distance as well as other unknowns. To

measure the absolute length, we attempted to extract it for the fully stretched head-hinge construct at 1.5 pN force. To this end, we applied the constant force of 1.5 and then moved the stage to the opposite direction until the force reached −1.5 pN and from this distance calculated the absolute length of the cohesin. Our results showed that the inferred length varied between 10 and 65 nm (Supplementary Fig. 6g) for the complex with expected length of ~50 nm. Therefore, we concluded that our technique does not allow us to determine the absolute length of static complexes.

The length of the coiled coil tether was extracted from electron microscopy measurements to be 110 nm (Supplementary Fig. 7a,b).

For the analysis of the head-hinge construct, only molecules that clearly showed transitions between fully bent and fully extended states were selected. Traces exhibiting partial bending or no bending were discarded due to uncertainty in attributing such molecules to one of the three states. Selected traces were further additionally aligned according to their length in the fully extended state and a cumulative histogram with a bin size of 1 nm was plotted for each value of the stretching force. Histograms were fitted with multiple Gaussian peaks using OriginPro 2021 and the center for each peak was evaluated.

## Fit of the head-hinge histogram data to a three-state model

For the three-state model shown on Fig. 1f the probability of finding the system in each of the states denoted by A (bent), B (intermediate) and C (extended) is given by:

$$\begin{cases} \frac{dA}{dt} = -k_1 A + k_{-1} B \\ \frac{dB}{dt} = k_1 A - k_{-1} B - k_2 B + k_{-2} C \\ \frac{dC}{dt} = k_2 B - k_{-2} C \end{cases} \qquad (2)$$

The probability of finding the system in each of the states in steady state can be compared to the relative lifetime spent in each state from Fig. 1d assuming sufficiently large number of transitions and molecules have been sampled.

Considering normalization, in the steady state this system of equations becomes:

$$\begin{cases} k_1 A = k_{-1} B \\ k_2 B = k_{-2} C \\ A + B + C = 1 \end{cases} \qquad (3)$$

The rate constants of the forward and reverse reactions are related to force as: $\frac{k_1}{k_{-1}} = K_0^{1,2} e^{\frac{\delta_1 F}{k_B T}}$ and $\frac{k_2}{k_{-2}} = K_0^{2,3} e^{\frac{\delta_2 F}{k_B T}}$, where $K_0^{1,2}$ and $K_0^{2,3}$ equilibrium constants for the transitions between bent and intermediate and intermediate and extended states correspondingly. The solution is given by:

$$A = \frac{1}{1 + \frac{k_1}{k_{-1}} + \frac{k_1 k_2}{k_{-1} k_{-2}}} = \frac{1}{1 + K_0^{1,2} e^{\frac{\delta_1 F}{k_B T}} (1 + K_0^{2,3} e^{\frac{\delta_2 F}{k_B T}})}; B = A \cdot K_0^{1,2} e^{\frac{\delta_1 F}{k_B T}}; C = B \cdot K_0^{2,3} e^{\frac{\delta_2 F}{k_B T}}$$

$$(4)$$

A, B and C as functions of forces are plotted as solid lines in Supplementary Fig. 1f with $K_0^{1,2}$ and $K_0^{2,3}$ as two independent fit parameters and $\delta_1 = 17$ nm and $\delta_2 = 15$ nm.

## Molecular dynamics simulation protocol

Template PDB files (PDBID: 5T8V, 6YUF) of crystal structures of open and gripping state of cohesin loader SCC2 from *Chaetomium thermophilum* and *Schizosaccharomyces pombe*, were obtained from Protein Data Bank. Models were generated by SWISS-MODEL[55] using the template PDB files and the sequence of *Schizosaccharomyces pombe* SCC2 (UNIPROT: Q04002). DNA-bound models were generated by superimposing the Cα positions of the modeled SCC2 apo structures to the

DNA-bound cryo-EM structure (PDBID: 6YUF) to dock the DNA. The structure models were solvated with SPC water model and neutralized with additional ions resulting in a final salt concentration of 40 mM similar to the salt concentration used in loop extrusion in vitro assays. The simulation box had 1 nm distance to the wall, and periodic boundary conditions were applied. Before simulations the system was subjected to the energy minimization of 50,000 steps. The simulation was carried out at constant pressure of 1 atm and constant temperature of 300 K.

Energy minimization and equilibration of the apo- and DNA-bound models were performed using GROMACS, with AMBER99SB force field.

The initial center of mass distance between the bent and relaxed NIPBL structure obtained by by SWISS-MODEL was 35.2 Å. After the energy minimization this distance did not change for the structure without DNA and decreased to ~26.4 Å in the presence of DNA. During the equilibration step this distance stayed almost the same for the structure without DNA and reduced further to ~18 Å for the structure with DNA (Supplementary Fig. 4).

To pull relaxed structure to the bent form we applied pulling force to each atom using a quadratic potential with stiffness kappa equal 300 kcal/mol/Å$^2$. For the umbrella sampling, 50 conformations of 40 ps intervals were sampled along the 2 ns trajectory. MD simulations were performed on all 50 windows for 2 ns and harmonic restraints on RMSD were applied to each window. The trajectories of 50 simulations were combined and all biases from each frame were extracted. WHAM analysis was done to extract the free energy change associated with each conformation in $k_B T \cdot \ln w$ (where, $w$ is bias potential). The free energy profiles along the trajectory were plotted against the COM distance between the N-terminal domain (residue 1–568) of the starting and end conformations when aligned by their C-terminus Cα positions (residue 569–1321). The COM distance was calculated using VMD[56]. Targeted molecular dynamics and Umbrella sampling were done using the PLUMED[57] plugin in GROMACS.

**Metropolis Monte-Carlo Simulations of Cohesin-DNA interaction**

To simulate how head-head disengagement can lead to formation of a DNA loop, we used the molecular-mechanical model we developed earlier[19]. Briefly, both DNA and Cohesin were described as dssWLC model[58]. DNA was defined as a sequence of beads with positions $r_i$ and an orientation unit vector $u_i$ attached to each bead. The DNA contour length was chosen 325 nm, the distance between beads 5 nm and DNA persistence length 50 nm for all simulations.

Cohesin was modeled using five beads representing the SMC1 and SMC3 heads, the SMC1 and SMC3 elbows as well as the hinge. Similar to DNA, each bead is defined by its position and orientation vectors (e.g., $r_{Smc3h}$, $u_{Smc3h}$ for the Smc3 head, etc.). All parameters were as in ref. 19.

The interaction between DNA and SMC3 was assumed to be slipping: DNA is free to diffuse along the surface of SMC3 subunit. This was described the same energy term as in ref. 19:

$$E_{Smc3-DNA} = \frac{\alpha}{2}|\mathbf{R}_\perp|^2 + \frac{\beta}{2}|\mathbf{n}|^2 \tag{5}$$

where $\mathbf{R}_\perp$ is the shortest distance between SMC3 and the center of the closest DNA bead,

$$\mathbf{n} = \mathbf{u}_{Smc3h} \times (\mathbf{r}_{Smc1h} - \mathbf{r}_{Smc3h})/|\mathbf{r}_{Smc1h} - \mathbf{r}_{Smc3h}| - \mathbf{u}_i, \tag{6}$$

where $i$ is the index of the DNA bead currently interacting with SMC3, and the second term is introduced to take into account the orientation of DNA with respect to the orientation of cohesin.

The interaction between SMC1 and DNA was assumed to be rigid – DNA cannot diffuse along SMC1, but must stay bound at the place of interaction. The corresponding energy term is simply:

$$E_{Smc1-DNA} = \frac{\gamma}{2}|\mathbf{r}_{Smc1h} - \mathbf{r}_i|^2 \tag{7}$$

where $\mathbf{r}_i$ is the position of the DNA bead to which SMC1 head is bound. All stiffnesses $\alpha, \beta$ and $\gamma$ were the same as in ref. 19 and all numerical calculations were carried out with the Metropolis method for Monte-Carlo simulation.

This model has no explicit chemical kinetics, and to simulate transitions between chemical states we prescribed parameter changes corresponding to a new state, then sampled a sufficient number of iterations to reach a new equilibrium. To simulate transition from the ATP-bound to the post-hydrolysis state, we imposed parameter changes on the equilibrium length between SMC3 and SMC1 heads. We then sampled conformations with the new parameters until a new equilibrium was reached. As heads disengage, DNA remains bound to the SMC1 head and we assumed that this interaction allows DNA to adopt any arbitrary angle, which changes as heads separate. At the same time we assumed that DNA binding at the SMC3 allows DNA to slide in one direction perpendicular to the head-hinge and cohesin plane. The code used for the molecular-mechanical simulation of behavior and DNA loop extrusion is available in the GitHub repository. Details of the DNA loop extrusion assay are provided in the Supplementary methods.

For the model presented on Fig. 4d, we specifically asked whether we could find a plausible mechanism in which SMC head disengagement powered by the energy released from NIPBL$^{Scc2}$ straightening could lead to bending of a straight piece of DNA associated with cohesin into a small loop. As a starting point, we assumed that DNA is straight, and it is positioned between the engaged SMC1 and SMC3 heads (Fig. 4d). Supported by the recent data we assumed that both SMC1 and SMC3 heads can bind DNA[24]. Naturally, if both SMC heads are bound to DNA in the engaged state, the head disengagement cannot proceed without external force, because the two heads are effectively holding onto the DNA (Supplementary Fig. 5a). However, when we assumed that unbending of NIPBL$^{Scc2}$ can generate force and push SMC heads apart effectively assisting disengagement, we identified conditions under which DNA can form a small loop (Fig. 4d, Supplementary Fig. 5b). Our simulations predict that DNA loop forms during head disengagement when SMC1 holds DNA strongly and pulls it away from SMC3-DNA binding interface, which allows DNA to slide at 60–90° angle with respect to direction of the head-head movement (Fig. 4d). Thus, our model predicts that loop initiation during head disengagement is possible if SMC3 and SMC1 cohesin ATPase heads have different DNA binding characteristics. This was based on the assumption that SMC1 binds DNA stronger then SMC3 and the latter allows DNA to slide in a specific orientation perpendicular to the head-head movement.

**Protein expression labeling and purification**

Human cohesin and NIPBL were expressed and purified as described[24]. Briefly, vectors containing SCC1(3xTEV)-Halo, 10His-STAG1, SMC1-AVItag, SMC3-ybbr-Flag and BirA or Strep-Flag-Halo-Nipbl-10His were produced in SF9 cells. 3xTEV site was positioned at the residue 471 of SCC1. Cohesin complexes were expressed in baculovirus-infected SF9 cells. For viruses expressing BirA, the growth medium was supplemented with 50 μM D-biotin (Sigma).

After purification, cohesin was labeled with 20 μM Haloligand-Coenzyme A via ybbr tag. Unreacted haloligand was removed by ultracentrifugation in 5–25% sucrose gradients.

For optical trapping experiments, cohesin was attached to a bead via ~100 nm long passive handle made of myosin V coiled-coil[59]. To

generate the handle, coiled-coil domain of rat myosin 7 (residues 1231 to 1935) was assembled with an N-terminal 6His- and Halo-tag and a C-terminal Avi-tag. The construct was expressed in BL21(DE3) cells and purified using Toyopearl resin followed by HiTrap Q-FF ion exchange column.

## Transmission electron microscopy imaging

Protein samples were diluted down to 50 nM concentration both for cohesin and the coiled-coil linker. 5 µL of diluted protein samples were loaded on a glow-discharged (45 mA, 30 s) Carbon film Copper grid (C400Cu, EMResolutions), incubated for 2 min before grid staining in 40 µL droplets of 2% uranyl acetate for 5, 10, 15, 20 s sequentially. The grid was then blotted dry and micrographs were taken under x30 k magnification on a JEOL JEM 1400-Flash (JOEL) electron microscope operating at 120 KeV.

## Reporting summary

Further information on research design is available in the Nature Portfolio Reporting Summary linked to this article.

## Data availability

Source data are provided with this paper. Raw data will be made available by the authors upon request. Source data are provided with this paper.

## Code availability

MATLAB code for Monte-Carlo simulations as well as GROMACS and PLUMED codes are publicly available at https://github.com/FrancisCrickInstitute/CohesinModel.

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

## Acknowledgements

We thank all our laboratory members for discussions and critical comments on the manuscript. We thank Alipasha Vaziri for supporting initial stages of this project and we are grateful to Benedikt W. Bauer for numerous conceptual discussions and generously generating and providing the proteins used in this study. We thank Raffaella Carzaniga and the EM STP at the Francis Crick Institute for their help and support in obtaining EM images. This work received funding from the Francis Crick Institute in London and the Research Institute of Molecular Pathology in Vienna (IMP). JMP received funding from Boehringer Ingelheim, the Austrian Research Promotion Agency (Headquarter grant FFG-878286), the European Research Council (ERC) under the European Union's Horizon 2020 research and innovation programme (grant agreements No 693949 and No 101020558), the Human Frontier Science Program (grant RGP0057/2018) and the Vienna Science and Technology Fund (grants LS14-009, VRG10-011 and LS19-029). MM received funding from the UK Medical Research Council (FC001750), Cancer Research UK (FC001750) and the Wellcome Trust (FC001750). JMP is an adjunct professor at the Medical University of Vienna. For the purpose of Open Access, the authors have applied a CC BY public copyright licence to any Author Accepted Manuscript version arising from this submission.

## Author contributions

G.P. and M.M. designed the research and performed experiments; L.C. performed molecular dynamic simulations; M.M. performed MMC simulations; J.M.P. and M.M. supervised research; M.M. wrote the manuscript with input from all authors.

## Funding

## Competing interests

The authors declare no competing interests.
