## [Peer Review File · Nature Communications]

Single cohesin molecules generate force by two distinct mechanismsREVIEWER COMMENTS

Reviewer #1 (Remarks to the Author):

In the manuscript by Pobegalov et al., the authors use optical tweezers to assess the force-extension-relationship along two axes of the cohesin molecule, identifying conformational changes with distinct responses to an external load force. These experiments are complemented by molecular dynamics simulations to compute potential stored energy in a conformational change of the NIPBL cohesin subunit. The authors conclude that the coiled coils of cohesin exist in three conformations and that the transitions between these conformations are governed by thermal (Brownian) motions. In contrast, the association of the two ATPase heads constitute a power stroke that presumably transfers and stores its energy in bending of the NIPBL subunit.

The question how SMC complexes use ATP hydrolysis to translocate on DNA is of interest to a wide audience and its resolution will require the experimental demonstration of how they can exert forces. I highly appreciate the authors' efforts to establish a system that can provide meaningful information about the cohesin reaction cycle and hence help to understand how SMC complexes perform their mechanochemical work. However, the current manuscript lacks essential control experiments, does not show the data for the controls mentioned in the text, contains misrepresented structural information, and falls short in connecting the proposed steps in the cohesin reaction cycle to its actual DNA substrate. As a consequence, the results included in this manuscript do not sufficiently support the main claims of the manuscript, namely that ATPase head engagement provides the power stroke for DNA loop extrusion and physical energy generated in this process is stored in a conformational change of NIPBL. I cannot recommend publication in Nature Communications, but would like to encourage the authors to consider the comments below for a future publication.

Major points:

1. The key conclusion reported in the manuscript is that ATP binding drives a conformational change (power stroke) that acts against high (15 pN) external forces. There are multiple problems with this claim:

a. Power strokes of motor proteins are generally referring to the steps that generate force along the tracks they move along (e.g., DNA, RNA, actin, tubulin) or processive rotation and torque (e.g., ATP synthase, flagellar motor). The authors describe a conformational change without a clear connection to

how this motion would be transferred to enact mechanical work in its biological environment, especially as the motion described is orthogonal to the orientation of DNA bound to the ATPase head domains.

b. Power strokes need to occur processively in order to perform work. The single steps against force described in the manuscript are counteracted by subsequent reverse steps. The step size of 10 nm is in the range of Brownian fluctuations of the bead system (Fig. S3b): from a single, reversing, step ($\Delta s = 0$), it is impossible to infer power generation.

c. Power strokes by most motor proteins are a result of transmission of nucleotide binding state into large conformational changes elsewhere in the protein. For ATP-binding cassette ATPases, to which cohesin belongs, two nucleotides are bound at the interface of the ATPase head domains, resulting in conformational changes elsewhere in the complex. Unless the authors propose that ATP binding at the SMC1 and SMC3 ATPase heads results in a conformational change in the coiled coils that actively brings the heads together, the ATPase heads first need to come together and sandwich the ATP before they can generate any forces.

d. Structural studies of cohesin (Petela et al., 2021; Bürmann et al., 2019) show that ATPase heads in the absence of ATP are very closely aligned. The transition to the ATP-bound state requires only 2 nm (not 10 nm) of movement. Petela et al., 2021 should be included at this point (lines 340-344). Proximity of the heads in the absence of ATP is also supported by smFRET efficiencies ~ 0.4 (Bauer et al., 2022). The large engagement step observed thus seems an artificial result from the exerted pulling force, which places the heads in a thermodynamically less favoured conformation.

Taken together, the transition described for the SMC ATPase heads falls short from the usual definitions of a power stroke as a “large, rapid structural change in a protein that can be used to do mechanical work” (Howard, 2006), and is rather a description of ATP binding and unbinding (see below) from an artificial, thermodynamically disfavoured conformation. Unless the authors can couple this transition to consecutive steps on DNA (for instance by measuring bead-bound SMC1 against surface-bound DNA), it cannot be called a power stroke and its relevance to the mechanochemical reaction cycle of cohesin remains unclear.

2. The authors present NIPBL as a mechanical spring that is loaded by the dimerization of the ATPase heads to store energy. While I agree that the “bent” conformation might have a free energy penalty associated with it, there is no evidence that this conformation is induced by the mechanical movement of the ATPase heads:

a. The authors omit the very important fact that the comparison of the two states of NIPBL as depicted in Fig. 3A-B are derived from structures with and without DNA as much as ATP being bound to the SMC heads. This is critical, since NIPBL in the various cryo-EM structures (Shi et al., 2020, Higashi et al., 2020 and Collier et al., 2020) makes extensive salt bridges with the DNA phosphate backbone. Disregarding other types of contacts to the DNA or other subunits, the combined free energy change of just these salt bridges falls in the same order as the calculated free energy change of the bent state, and the establishment of these contacts requires NIPBL to curve around the DNA.

b. If the bending of NIPBL were coupled to the SMC ATPase engagement and disengagement cycle, one would need to assume that the contacts between NIPBL and the SMC subunits are maintained between the two conformations. Although the contact between NIPBL and SMC3 has not been resolved in the post-hydrolysis state (Fig. 3A), structures of the homologous condensin subunits (Lee et al., 2020) reveal distinct modes of binding of the NIPBL homolog Ycs4 to the SMC3 homolog SMC2 between ATP-bound and bridged states. In other words, the contact between NIPBL and SMC3 is most likely lost during the conformational transition, making it difficult to imagine how it could act as a spring.

c. If the NIPBL spring were to store the 18 kcal/mol energy provided by head dimerization, and the power stroke supposedly delivers 22 kcal/mol (in the presence of NIPBL), the total energy expenditure in the experiments would be 22+18 kcal/mol (displacement against the bead force and the NIPBL spring), which substantially exceeds the hydrolysis energy provided by 2 ATP molecules, <30 kcal/mol (lines 303-310). The parameters presented do not add up to a thermodynamically possible course of events.

The proposed mechanical coupling of head-head engagement with compression of a NIPBL-loaded spring are thus not supported by the physical parameters presented by the authors and result at least to a very significant extent from ignoring the bound DNA.

Specific comments:

1. In Fig. 1C-D, the authors only show data for a single condition (+NIPBL+ATP+DNA) for the conformational changes of the SMC coiled coils. It is unclear whether these changes are dependent on ATP or DNA. The authors should compare the conformational dynamics without DNA or without ATP, as well as in a locked ATP-bound state (such as a hydrolysis-deficient mutant or using non-hydrolysable nucleotide analogues). Rate constants for each state transition should be determined and presented. Whenever conditions were tested, the authors need to provide the data. Even from non-dynamic molecules, extension and relaxation experiments can still provide meaningful information.

2. For the SCC1 TEV cleavage experiments in Fig. 1E, the authors need to specify where the TEV site is located. The integrity of the cleaved complexes should be demonstrated (for example by size-exclusion chromatography and SDS-PAGE to demonstrate that these complexes still contain all components, including NIPBL). The proposed model can be more specifically addressed by lengthening or shortening SCC1 and demonstrating a corresponding change in the intermediate state, or by identifying and mutating the SCC1 residues involved in the proposed contacts.

3. The schematics in Fig. 1F should make it explicit that the experiment provides no information on the relative position of the second, non-trapped (SMC1) head. This should be indicated in the text (line 160). Bending of the coiled coils are observed in cryo-EM structures at both the joint and elbow, which should at least be outlined in the text.

4. The experiments in Fig. 2B lack a comparison of +NIPBL-ATP and -NIPBL -ATP conditions, as well as the dependence on DNA. These data need to be included.

5. The disengagement rate plotted in Fig. 2E is a combination of disengagement of the SMC1:2ATP:SMC3 module (k_{off}), the rate of ATP hydrolysis, and the rates of Pi and ADP release. Some of these aspects can be dissected by repeating the experiments with mutant cohesin complexes incapable of hydrolysing ATP.

6. The SMC head-head interaction is a simple binding event, where the binding probability (and k_{on}) at constant ATP concentrations depend on the collision of the two heads, which in the setup depends directly on the movements of the beads attached to one of them ($\langle x^2 \rangle = kT / \kappa$) and keeping them an average of 10 nm apart, and thus scales with the trap stiffness $\sqrt{kT / \kappa}$, rather than the force exerted, $e^{(\delta F / kT)}$. Because the authors change the trap stiffness whenever the pulling force is altered, the engagement rate gradually drops (Fig. 2F). This can be easily demonstrated by repeating the experiments in Fig. 2F at the same trap stiffness.

7. Along these lines, removal of NIPBL and its contribution to restraining the extension of the heads at 15 pN, results in an average spacing of the heads 20 nm apart, which becomes hard to overcome by the bead fluctuations.

8. The authors should assess the dependence of the head engagement rate on ATP concentrations to discern whether ATP molecules need to bind simultaneously or sequentially.

9. To demonstrate that the 10-nm step indeed measures ATPase head disengagement, the authors could make use of a recent construct that allows locking the ATPase heads together by addition of rapamycin (cohesin SMC1-FKBP/SMC3-FRB; Bauer et al., 2021).

10. The structure models depicted in Fig. 3A-B need to show DNA whenever there is actually DNA present in the structures. I also recommend including a DNA molecule in the molecular dynamics calculations to determine its contribution to stabilising the 'bent' state.

11. The coiled-coil handle loop extrusion experiment shown in Fig. S3G lacks statistics for the observed frequency of loop extrusion as well as controls that ensure the observed activity can be attributed to the labelled population. Either the labelled protein should be separated from unlabelled (e.g., by size-exclusion chromatography), or a fluorophore should be included in the handle and co-imaged with the DNA.

12. The references to the Lee et al., 2020 and Collier et al., in lines 73 and 75 seem to be swapped.

13. Contrary to what the authors state (lines 75-80), the smFRET data from Bauer et al., 2021 shows that the hinge-head distance is an ensemble irrespective of the presence NIPBL or ATP. Moreover, cryo-EM structures of cohesin in the absence of ATP also show the hinge in the vicinity of the SMC3 head (Petela et al., 2021). This section is a misrepresentation of the available structural evidence.

14. It should be possible to establish at which distance the static state is fixed in the absence of NIPBL from force-extension measurements and this data should be included (lines 121-122).

Reviewer #2 (Remarks to the Author):

Cohesin is a member of the SMC family of proteins and plays crucial roles in organization of the 3D genome in cellular functions. In this manuscript, Pobegalov et al. used optical trapping, combined with computational methods, to investigate conformational dynamics of cohesin.

Overall, the authors presented solid evidence for the 3-state, thermally driven, coiled-coil bending of the cohesin arm. Arguments based on the observed 10 nm engagement/disengagement step of the ATPase heads and its energetic implications, as shown in molecular dynamics simulations of relaxing a bent NPIBL subunit, are compelling. Based on these findings, the authors proposed a model in which energy

from the engagement of ATPase heads is stored in the bent NIPBL subunit and its release allows the protein to overcome the activation barrier in local DNA bending when initiating loop extrusion. The text is clearly written and flows logically throughout.

The following, many minor, points should be addressed/clarified regarding some specific statements in the text.

1. In Fig. 1D, the peak centers of histogram at 0.5 pN are separated by a distance of >20 nm between and completely folded and intermediate states. Can the authors clarify this discrepancy compared to the stated 17 nm step (as marked by grey dashed lines)?
2. In discussing the experiment involving TEV-cleavage of SCC1 leading to a static cohesin molecule under 1 pN force (Fig. 1E), the state is assumed to be fully extended (Line 174). Can the authors explain how the particular state is assigned over the other two? As the authors state in the Methods section (Lines 601-603), traces that were static would have the uncertainty in state assignment?
3. Controls:
 - a. Regarding the experiment shown in Fig. 1E, TEV cleavage is assumed to have taken place, which in turn resulted in a change in cohesin behavior under force. While not unreasonable, it would be more convincing to show that under the experimental condition used (introducing TEV after immobilizing cohesin), TEV did indeed cleave and not just merely affect the force behavior (i.e. make cohesin arms static) some other way.
 - b. In Supp Fig 3G, the authors show that CC-handle coupled cohesin is active in a DNA looping assay using surface tethered DNA and cohesin in solution. Considering that the configuration of optical trapping experiments used surface-tethered cohesin, it would be better to show activity with that particular attachment geometry, or at least explain in the Methods why the control is carried out somewhat differently.
4. In Line 143, the authors stated that the angle formed at the hinge in the intermediate state is ~60 deg. It would be helpful to briefly explain in the Methods how that is calculated.
5. Compared to head-hinge movement measurements, head-head engagement dynamic experiments were carried out in the absence of DNA. Can the authors comment on what specifically led to the decision of not including DNA in those experiments?
6. In Fig. 2B, under the +NIPBL condition, did the author observe a rip of ~10 nm in the force ramp up to 20 pN? Not sure why the force-extension plot is omitted for force < 10 pN. It could be more interesting to show an example of a rip/head disengagement in this plot, if indeed it did happen on a regular basis, in the presence of ATP and NIPBL.
7. Regarding experiments presented in Fig. 2C, what percentage of molecules showed 10 nm transitions at forces <20 pN in the presence of NIPBL and ATP? (Line 213-215). In addition, it could be helpful to show an example trace at 15 pN in 2C, in addition to 5 pN and 10 pN, since it is stated that the transitions occur up to 15 pN (Lines 221-222).

8. Regarding Fig. 2D, can the authors comment on if there was any correlation between force load and observed step size? Or correlation between an engagement distance vs. the following disengagement distance (or the other way around)?
9. The authors stated that in the absence of ATP, there was no observed engagement transitions under force (Lines 251-252). Have the authors tried varying ATP concentrations to see if rates of engagement or disengagement depended on ATP concentration? Similarly, have the authors tried nonhydrolyzable ATP analogues? Would the authors expect it to affect disengagement rate? Would blocking hydrolysis strengthen the argument that the head engagement itself is not dependent on hydrolysis and/or that disengagement is post-hydrolysis?
10. Line 598 in the Methods section refers to “Supplementary Figure E, F”. Presumably it should be “Supplementary Figure 3 E, F”.
11. Legend for Fig. 1C (Line 927), 1D (Line 931), 1E (Line 933), and Fig. 2B (Line 940). These corresponding figure panels have axes labeled “relative distance/extension”. The text in figure legend does not specify “relative”.
12. Legend for Fig. 3B, Line 954, “Bend” should be “Bent”. In the following sentence, the red appears to represent “centre of mass distance” rather than ‘centre of mass” as stated?

Reviewer #3 (Remarks to the Author):

Pobegalov et al. use optical tweezers to examine the energetics and dynamics of cohesin complexes in a surface-tethered assay. They observe bending of the cohesin hinge with an intermediate state and describe its force dependency with a 3 state model. Furthermore, they use a similar assay to monitor cohesin-head-head dynamics, which they describe as a power stroke able to generate up to 15 pN in force. Using MD simulations, they find that a part of this mechanical energy is stored in the conformation of NIPBL-Scs2. They discuss these results in the context of the mechanism of loop extrusion with the further support by MC simulations. They speculate that the two different molecular movements they observed in their experiments play distinct roles in loop initiation and loop elongation.

The manuscript is very exciting. It provides clear mechanistic insight into this very relevant topic. The research is designed well and conducted carefully, the findings are very interesting and well interpreted, and the paper is written very well. The findings will be of interest to the readers of this journal.

I have a few minor remarks/questions:

- It would be helpful for the assessment of the data if the standard deviation of the force during force clamps would be given or sample traces would be shown in the SI.
- Fit parameters for Suppl Fig 1F are not given, and would be potentially interesting
- The data in Fig 2B is described in the text as elastic with a constant stiffness of 0.7 pN/nm, however, the data and the not further specified trend line both look rather non-linear.

- The exponential fit in 2F with a fixed length change is slightly misleading since commonly this parameter is found to be smaller than the associated structural change. Also, the (linear?) trendline is not very helpful as an exponential dependency is expected. Maybe a better approach would be to perform an exponential fit with the length change as a fit parameter. I would expect the best fit length change to be much smaller than 10 nm, which could then be discussed. I nevertheless agree with the authors that the weak force dependency and the head engagement happening even at 15 pN favors a power stroke mechanism over a Brownian ratchet mechanism.

- The authors compare the energy they estimate for the power stroke with the elastic bending energy of NIPBL-Scc2. However, they measured the power stroke energy in the presence of NIPBL-Scc2. I think this needs to be considered, since the power stroke needs to overcome both the force applied by the optical trap as well as the bending energy of NIPBL-Scc2. This can also be seen in figure 4B, where the lower line should presumably read $\Delta G = \Delta G (\text{ATP binding}) - \Delta G (\text{Scc2 bending})$.

- Page 8 "Thus, our data combined with previous observations suggest that the energy that leads to the force generation during the head engagement arises from ATP binding, but not hydrolysis." This is not entirely clear to me, the authors could maybe elaborate on this argument.

- It is not entirely clear to me where the directionality of loop elongation by the head-hinge-movement would come from, especially if it is entirely driven by thermal motion of the hinge (Fig. 4E). This might go beyond the scope of this manuscript but might be worth discussing.

- The way the data is analysed is very sound. However, I wonder what the impact of the other geometrical parameters in Equation (1) is. How was the distance of the bead from the flow cell adjusted to 60 nm and what is the uncertainty of this adjustment? Similarly, how was the bead diameter determined and how would a deviation from the assumed bead diameter impact the data?

Small details:

- Page 7, line 330: comma after however is missing

- Page 2, line 117: If the authors could give concentrations here, it would be rather helpful to the reader.

Reviewer #4 (Remarks to the Author):

The manuscript by Pobegalov et al is a timely and important contribution as it provides measurements critical for understanding SMC loop extrusion dynamics. Importantly, these optical trap measurements under external force are complementary to dynamic AFM and FRET data, providing a wealth of new information and leading to new conclusions. Those include: (i) an existence of an intermediate state; (ii) largely fluctuation driven head-to-hinge dynamics; (iii) generation of a very considerable head-to-head force by ATP binding; (iv) potential storage of the mechanical energy in the deformation of Nipbl

molecule. While the manuscript is generally convincing, I have doubts about some conclusions that I challenge below.

In general, I enjoyed reading this very solid and thought-provoking study and I only wish authors were to present more data such as data for different combinations of ATP/DNA/Nipbl (that I'm sure have been generated) and for non-hydrolyzable ATP, that can only strengthen their conclusions.

1. Head-to-hinge movement.

Authors imply that the bent and intermediate state at 17nm separation corresponds to the "elbow-bent" states, but observation of the O-looking state for other SMCs in dynamic AFMs may suggest some other conformations. I just suggest authors mention this as a possibility.

Does the state of the intermediate (i.e. 17nm) depend on the force? It appears that 0.5pN force shows a peak at around 25nm instead. How does this state depend on the presence of ATP/DNA/Nipbl?

Authors state that in the absence of ATP/DNA/Nipbl and at high salt the molecule is largely static, but don't show any data, so it's difficult to understand what is this static state (extended? bent?). What would be the distribution of states at zero force? at physiological 150mM salt?

In fact the model used to fit the data has two adjustable parameters that represent occupancy of the intermediate and bent states at zero force. Reporting these values and comparing with zero force data (if any) could be very instructive.

I have some doubts in the strength of the main conclusion about entirely thermal motion that drives head-to-hinge dynamics. What the data show is that the system doesn't generate a substantial (>1pN) force on contraction, but I'm not sure one can rule out fast and infrequent power-strokes on expansion. If fast, such movements would not significantly affect statistics of states for the rest of dynamics.

Also if dynamics were entirely fluctuation driven it would not get affected by ATP hydrolysis and thus would be similar to dynamics of the system when a non-hydrolyzable ATP were used. It would be great to see data for such experiments.

2. Head-to-head dynamics

I am concerned about the total work measured $\sim 150\text{pN nm}$, upon head-to-head engagement as it significantly overshoots the energy of ATP hydrolysis that is $\sim 10kT = 40\text{pN nm}$.

<http://book.bionumbers.org/how-much-energy-is-released-in-atp-hydrolysis/>

For two ATP that gives 80 pN nm , but still way below 150 pN nm .

Authors suggest that the main energy source is ATP bindings rather than hydrolysis or release. This hypothesis can be more directly tested by (i) varying ATP concentration as it would determine the energy gained by binding; (ii) using non-hydrolyzable ATP or SMC mutants that don't hydrolyze ATP.

Would be great to see the role of DNA in this process. I couldn't find data for +/- DNA, (and not for some other conditions among +/- Nipbl, +/- ATP)

Storage of energy in Nipbl deformation is plausible, modulus that the total work required can be too large as indicated above. Simulations suggesting the role of head-to-head movement in initiation of extrusion are certainly thought-provoking. I couldn't follow how such a movement can assist in loop extrusion at stages past the initiation -- authors may want to elaborate on this or just explain the initiation is the key step where this movement plays a role.

A very nice paper!

Leonid Mirny

Remark to all reviewers

We would like to thank the reviewers for their critical reading of our manuscript. We were delighted to see some extremely positive and enthusiastic assessment of the reviewers, who recognized the innovative aspects of our work and considered our manuscript “compelling”, “exciting”, “providing clear mechanistic insight into this very relevant topic”, “designed well and conducted carefully”, “clearly written and flowing logically throughout”.

We also found the referees’ constructive suggestions very valuable as they helped to significantly improve the quality of our manuscript. Before providing our point-by-point response, we would like to briefly summarize the main implemented improvements and additions:

1. We have performed multiple additional control experiments that addressed the separate roles of ATP, NIPBL and DNA in the dynamics of the head-hinge movement and head-head engagement, which confirmed our previous findings.
2. We have performed additional experiments to investigate the dynamics of both, head-hinge and head-head movements in the presence of non-hydrolysable ATP analogue AMPPNP and found that it can support dynamics of the head-hinge movement, but not the head-head engagement.
3. We have performed additional MD simulations of the NIPBL bending in the presence and absence of DNA. Our new findings show how DNA may facilitate NIPBL bending and supports conclusions of our new experiments.

We have revised Figures 1 – 3 and added additional Supplementary Figures S1 – S7 as well as made essential corrections and extensions to the original main text of the manuscript (see unlined changes in the revised manuscript). In addition, we addressed all the comments from all the reviewers, as detailed in our point-by-point response below.

REVIEWER COMMENTS

Reviewer #1 (Remarks to the Author):

In the manuscript by Pobegalov et al., the authors use optical tweezers to assess the force-extension-relationship along two axes of the cohesin molecule, identifying conformational changes with distinct responses to an external load force. These experiments are complemented by molecular dynamics simulations to compute potential stored energy in a conformational change of the NIPBL cohesin subunit. The authors conclude that the coiled coils of cohesin exist in three conformations and that the transitions between these conformations are governed by thermal (Brownian) motions. In contrast, the association of the two ATPase heads constitute a power stroke that presumably transfers and stores its energy in bending of the NIPBL subunit.

The question how SMC complexes use ATP hydrolysis to translocate on DNA is of interest to a wide audience and its resolution will require the experimental demonstration of how they can exert forces. I highly appreciate the authors' efforts to establish a system that can provide meaningful information about the cohesin reaction cycle and hence help to understand how SMC complexes perform their mechanochemical work. However, the current manuscript lacks essential control experiments, does not show the data for the controls mentioned in the text, contains misrepresented structural information, and falls short in connecting the proposed steps in the cohesin reaction cycle to its actual DNA substrate. As a consequence, the results included in this manuscript do not sufficiently support the main claims of the manuscript, namely that ATPase head engagement provides the power stroke for DNA loop extrusion and physical energy generated in this process is stored in a conformational change of NIPBL. I cannot recommend publication in Nature Communications, but would like to encourage the authors to consider the comments below for a future publication.

We thank the reviewer for acknowledging our efforts to shed new light on this topic. In the revised manuscript we have performed additional controls requested, which address concerns raised by the reviewer. We also apologize that some of the misunderstanding might have been due to poor explanation from our side, which we now corrected in the revised version.

Major points:

1. The key conclusion reported in the manuscript is that ATP binding drives a conformational change (power stroke) that acts against high (15 pN) external forces. There are multiple problems with this claim:

a. Power strokes of motor proteins are generally referring to the steps that generate force along the tracks they move along (e.g., DNA, RNA, actin, tubulin) or processive rotation and torque (e.g., ATP synthase, flagellar motor). The authors describe a conformational change without a clear connection to how this motion would be transferred to enact mechanical work in its biological environment, especially as the motion described is orthogonal to the orientation of DNA bound to the ATPase head domains.

We thank the reviewer for this comment. Our data presented in the manuscript show how two conformational changes in cohesin molecules (head-hinge and head-head) generate mechanical force. Our main finding is that the force generated by the head-hinge conformational change is small because it is likely driven by thermal fluctuations, and the force generated by the head-head movement is significantly larger, likely because it uses energy of ATP binding. We agree with the reviewer that how these two types of motion are used to power cohesin-DNA interaction such as for

example translocation, loop extrusion or DNA loading/unloading are not yet known. As we discuss below, answering these questions would require using different kinds of tools that would allow probing the coupling between the conformational changes in cohesin and its movement on DNA, which went beyond the scope of this study. However, the finding that two different large-scale conformational changes within one molecule can generate different amounts of force by different mechanisms is surprising and unusual. To our knowledge this is the first molecule that can do this and therefore, we believe it represents significant interest on its own.

We agree with the reviewer that it wasn't sufficiently clear in our original manuscript that we study ability to generate force by cohesin without directly showing how this movement is coupled to DNA. As reviewer correctly points out, there are power strokes generated by processive motors (the best studied example is probably kinesin-1 [e.g. <https://doi.org/10.1038/35036345>]). However, there are also power strokes in non-processive motors - for example in non-processive muscle myosin, which binds actin, changes the angle of its stalk in a conformational change which generates power stroke, and then detaches [e.g. <https://doi.org/10.1002/cm.10014>]. In the last case the power stroke is stimulated by actin, but it occurs regardless of whether there is load or cargo movement and no processive movement. In this case, generating the power stroke is the inherent property of the conformational change in the molecule that can exert force on external objects and it is not related to its ability to move processively. We propose that the conformational changes in cohesin works in the same way, and we have made this clearer in the revised manuscript.

b. Power strokes need to occur processively in order to perform work. The single steps against force described in the manuscript are counteracted by subsequent reverse steps. The step size of 10 nm is in the range of Brownian fluctuations of the bead system (Fig. S3b): from a single, reversing, step ($\Delta s = 0$), it is impossible to infer power generation.

An example of a well-studied non-processive motor that generates power stroke is a muscle myosin that we referred to above. This illustrates that power strokes do not necessary lead to the processive motion.

Figure S3b (now Supplementary Fig. 6b) shows movement of a bead without external force tracked by a video camera. We used it to select for tethers which were attached to the surface by only one cohesin, but as reviewer correctly points out, video tracking does not have sufficient resolution to detect conformational changes in the molecule. After the single tether was identified, we applied force using the laser and tracked the position of the bead with a quadrant photodetector, which has sub-nanometer accuracy. As we now show in the revised manuscript the resolution of our experiments increases with force because applied tension restricts Brownian movement. At 1 pN force our resolution is ~ 4.5 nm and at 5 pN already 2.5 nm (New Supplementary Fig. 1b). Our ability to resolve steps is also demonstrated by the Fig. 1C, Fig. 2C, Supplementary Fig. S2a, S3b and S3c, which show that the steps are clearly seen on top of the noise in these traces. Thus, we show that we can clearly resolve 10 nm steps in our assay.

The reviewer is correct that theoretically a single step in head-head engagement may be counteracted by the reverse step to cancel out the force generation. We consider this possibility in our manuscript and rule it out based on the data presented on new Figure 2f,g. Our data show that ATPase head engagement occurs in 10 nm abrupt step against external forces of up to 15 pN. There are only two different physical mechanisms that can explain it in principle. It must either be a two-state system, which can oscillate between two semi-stable states separated by ~ 10 nm at high force and as suggested by the reviewer not generate much force, or alternatively, it could be a power stroke which uses ATP energy to engage ATPase heads against the external force. To distinguish

between these two possibilities, we have measured engagement and disengagement rates and showed that they only weakly depend on the external force inconsistent with the first mechanism. Therefore, assumption that it is a two-state system that oscillates between the states can be rejected as it would require very strong exponential dependence of the transition rates on the external force as we now show in Figure 2f,g. Thus, we must conclude that the only physical possibility left that is consistent with our data is that the head-head movement is a power stroke.

c. Power strokes by most motor proteins are a result of transmission of nucleotide binding state into large conformational changes elsewhere in the protein. For ATP-binding cassette ATPases, to which cohesin belongs, two nucleotides are bound at the interface of the ATPase head domains, resulting in conformational changes elsewhere in the complex. Unless the authors propose that ATP binding at the SMC1 and SMC3 ATPase heads results in a conformational change in the coiled coils that actively brings the heads together, the ATPase heads first need to come together and sandwich the ATP before they can generate any forces.

We agree with the reviewer that one of the possibilities for the force generation could be that ATP binding induces a conformational change in the coiled-coils. However, our data show that coiled-coils are rather flexible and unable to support force generation above ~ 1 pN. There is another possibility not considered by the reviewer that we propose in our manuscript. It can be best illustrated on the example of a cohesin related protein - ABC transporter. Similarly, to what we propose for cohesin, it has been suggested that in ABC transporters, ATP binding to ATPase heads generates power stroke, which engages their heads [e.g. <https://doi.org/10.1021/jp507930e>]. The reviewer is correct that for the power stroke to be generated and transmitted into the head engagement movement, the conformational change needs to be generated against a mechanical scaffold which would be strong enough to allow the respective movement of the heads against the external force. In case of ABC transporters such mechanical scaffold is the transmembrane domain of the protein, which is stiff enough to support the power stroke and the resulting engagement of the ATPase heads. Our data show that in case of cohesin coiled coils are too flexible to provide the necessary mechanical support. We propose that it is instead provided by NIPBL. Several lines of evidence suggest this possibility. First, we show that the presence of NIPBL is absolutely required for the head-head engagement against external force in 10 nm steps and does not happen without it (New Figure 2e and S3b). Second, mechanical stiffness of the head-head linkage is increased by a factor of at least 3x in the presence of NIPBL and makes the head-head interface of the molecule essentially non-stretchable (Figure 2b). Thus, we showed that NIPBL significantly increases the stiffness between ATPase heads, allows ATPase heads to engage against forces up to 15 pN.

Our interpretation of these results is that ATP binding to ATPase heads makes the head separated state highly energetically unfavourable. We propose that ATP binding changes the interaction between NIPBL and ATPase heads and the following conformational change coupled with NIPBL bending brings the heads together to a new energy minimum state in which both heads are engaged and stabilized by NIPBL. The energy difference between the new state and the old state (no ATP bound, heads disengaged) provides free energy that can be used by this stroke to perform mechanical work (i.e. exert force on external objects such as optically trapped bead). In other words, cohesin action following the ATP binding can be described as a conformational change, which can generate mechanical work to which we refer as a power stroke. This is schematically illustrated on Figure 1 below. We have made this clearer in the revised manuscript.

Figure 1. Free energy diagram of the head engagement. ATP binding to cohesin with disengaged heads (start top right) leads to the head engagements, which has lower free energy by the amount of ΔG . This free energy difference can be used to produce external mechanical work coupled with the head engagement. After ATP hydrolysis, ATP release and head disengagement the molecule is ready to bind another pair of ATPs. This brings free energy another step down by the same amount ΔG and another cycle of the mechanical work can be performed. This cycle is repeated, and each highly energetic ATP binding event can generate free energy, which can be used to perform mechanical work. Thin arrows show the sequence of events. Black line is the potential energy landscape between engaged and disengaged states.

d. Structural studies of cohesin (Petela et al., 2021; Bürmann et al., 2019) show that ATPase heads in the absence of ATP are very closely aligned. The transition to the ATP-bound state requires only 2 nm (not 10 nm) of movement. Petela et al., 2021 should be included at this point (lines 340-344). Proximity of the heads in the absence of ATP is also supported by smFRET efficiencies ~ 0.4 (Bauer et al., 2022). The large engagement step observed thus seems an artificial result from the exerted pulling force, which places the heads in a thermodynamically less favoured conformation.

The reviewer is correct that the structure reported in Petela et al., 2021 shows two ATPase domains closely aligned. However, these results were obtained after crosslinking of cohesin interfaces prior to imaging using BMOE (Petela et al., CryoEM) or BS3 (Bürmann et al., Negative stain EM). Moreover, these studies report on incomplete cohesin structures missing either SCC3 (Petela et al.) or SCC2 loader (Bürmann et al) subunits, both of which are essential for cohesin's motor activity and can be directly involved in regulation of the heads separation as we show.

Contrary to the second part of the reviewer's statement, close alignment of ATPase heads in no nucleotide state contradicts data from Bauer et al., 2021. We could not find the FRET value of 0.4

cited by the reviewer in that manuscript. Quite on opposite, Figure 3B in Bauer et al., 2022 shows a typical smFRET trace in which head separation results in FRET signal value close to zero indicating a physical distance of over 8 nm [e.g. <https://doi.org/10.1186/1477-3155-11-S1-S2>]. Text on the page 6 of Bauer et al., 2021 says: “majority of transitions occurred between FRET efficiency values (E-values) close to $E=0$ and $E=0.8$ (Figure 3D)” and “Analyses of these complexes revealed low FRET in the absence of either NIPBL or ATP”. These data are fully consistent and are in excellent agreement with our power stroke measurement of 10 nm. Moreover, single-molecule imaging of cohesin complexes using high-speed AFM (Figure S2I, Bauer et al., 2022) also directly showed that the ATPase heads of cohesin dynamically move relative to each other and on average are separated by a distance of 10 nm, which is also in a good agreement with our measurements.

Taken together, the transition described for the SMC ATPase heads falls short from the usual definitions of a power stroke as a “large, rapid structural change in a protein that can be used to do mechanical work” (Howard, 2006), and is rather a description of ATP binding and unbinding (see below) from an artificial, thermodynamically disfavoured conformation. Unless the authors can couple this transition to consecutive steps on DNA (for instance by measuring bead-bound SMC1 against surface-bound DNA), it cannot be called a power stroke and its relevance to the mechanochemical reaction cycle of cohesin remains unclear.

We agree with the reviewer that the power stroke is a large conformational change that can perform mechanical work. Our data show that the size of the head-head conformational change is ~ 10 nm, consistent with data from Bauer, 2021 and we show that it can indeed generate external mechanical work by pulling the bead out of the trap. In the revised text we discuss that there are only two possible physical mechanisms that can explain head-head engagement observed in our experiments and consistent with Bauer et. al., 2021. Either head-head engagement generates power stroke or as the reviewer suggests head-to-head distance can change due to oscillations between “thermodynamically disfavoured conformations”. (Howard, 2006) proposes an elegant test to distinguish between these two possibilities. Rates of oscillations between thermodynamically disfavoured conformations should depend exponentially on force, whereas transitions due to power stroke should have little dependence on the external force. We have performed this test and our data unambiguously show that the rates of transitions between head-head movements cannot be accounted for by exponential dependence (Figure 2f,g). Thus, we must conclude that the only remaining possibility left is that the head-head movement generates power stroke according to a definition which satisfies all criteria in (Howard, 2006).

We agree with the reviewer that the next important question is whether the power stroke that cohesin can generate is indeed coupled to consecutive steps when it moves on DNA. As the reviewer suggests, answering this question would require monitoring the power stroke as cohesin moves along DNA potentially in loop extrusion assay. This would require development of new techniques and approaches that will combine loop extrusion assay with the measurement of the conformational changes in cohesin and are technically challenging. Thus, our current study shows that conformational changes in cohesin can generate mechanical force. We have made this clear in the revised manuscript.

2. The authors present NIPBL as a mechanical spring that is loaded by the dimerization of the ATPase heads to store energy. While I agree that the “bent” conformation might have a free energy penalty associated with it, there is no evidence that this conformation is induced by the mechanical movement of the ATPase heads:

We are delighted that the reviewer agrees that “the bent NIPBL conformation might have a free energy penalty associated with it”. As discussed in more details below in the response to the specific points we added new results that more clearly explain our interpretation of the relationship between the NIPBL bending and the movement of the ATPase heads. Our data show that bending of NIPBL into highly energetic state and the movement of ATPase heads are likely coordinated: NIPBL is bent in ATP bound state in which heads are engaged and relaxed when the heads are disengaged. We agree with the reviewer that our original interpretation that NIPBL bending is caused by the movement of ATPase heads might have been confusing. Rather, our results indicate that the two movements are likely aligned in time: NIPBL can only bend after ATP binds. Our interpretation is that after ATP binding the whole system approaches the new stable state with heads engaged and NIPBL bent. We have modified the revised manuscript accordingly to reflect this interpretation.

a. The authors omit the very important fact that the comparison of the two states of NIPBL as depicted in Fig. 3A-B are derived from structures with and without DNA as much as ATP being bound to the SMC heads. This is critical, since NIPBL in the various cryo-EM structures (Shi et al., 2020, Higashi et al., 2020 and Collier et al., 2020) makes extensive salt bridges with the DNA phosphate backbone. Disregarding other types of contacts to the DNA or other subunits, the combined free energy change of just these salt bridges falls in the same order as the calculated free energy change of the bent state, and the establishment of these contacts requires NIPBL to curve around the DNA.

We thank the reviewer for pointing this out. Indeed, in the original version of the manuscript the change in the free energy associated with NIPBL bending was calculated in the absence of DNA. In the revised manuscript we have performed additional calculation of the energy change now in the presence of DNA. Our new data showed that the direction of the free energy change remained the same as it was without DNA – bend NIPBL plus DNA is still much more highly energetic state comparing to the relaxed NIPBL. Thus, qualitatively the trend in the energy change in the presence of DNA appeared similar as without DNA. However, the absolute energy change was smaller in the presence of DNA (New Figure 3e,f). We concluded that bending of NIPBL has “free energy penalty associated with it” with or without DNA, but the bending NIPBL in the presence of DNA is more energetically favourable.

b. If the bending of NIPBL were coupled to the SMC ATPase engagement and disengagement cycle, one would need to assume that the contacts between NIPBL and the SMC subunits are maintained between the two conformations. Although the contact between NIPBL and SMC3 has not been resolved in the post-hydrolysis state (Fig. 3A), structures of the homologous condensin subunits (Lee et al., 2020) reveal distinct modes of binding of the NIPBL homolog Ycs4 to the SMC3 homolog SMC2 between ATP-bound and bridged states. In other words, the contact between NIPBL and SMC3 is most likely lost during the conformational transition, making it difficult to imagine how it could act as a spring.

We agree with the reviewer that how NIPBL contacts SMC3 in the post-hydrolysis state is so far unknown. Our data show that NIPBL is required for the head-head movement and force generation as detected in the optical trap (Revised Figure 2e and New Supplementary Figure S3b). Consistent with this the head-head movement as detected by smFRET also requires NIPBL (Bauer, 2021).

While the situation for cohesin is less clear, Figure 1d in (Lee et al., 2020) referenced by the reviewer shows apo-bridged state for condensin in which Ycs4 indeed contacts both SMC heads in the post-hydrolysis state consistent with our interpretation. This raises the possibility that NIPBL may indeed maintain contacts with both ATPase heads in ATP bound and post-hydrolysis states. This is indeed consistent with our data. We show that in the presence of the NIPBL, mechanical transitions between the engaged and disengaged states do not exceed 10 nm distance. We also show that

forces larger than 20 pN applied in the presence of NIPBL led to the second abrupt step that increased the head-head distance further to ~ 30 nm (Supplementary Fig. S6d). Additionally, when we applied force between heads without NIPBL we found that they can stretch to a similar amount of ~ 30 nm at similar forces (Fig. 2b). Thus, without NIPBL the distance between heads can open to about 30 nm. Since we only see 10 nm separation at forces lower than 20 pN and no further increase in the distance between them, this suggests that NIPBL likely maintains contact with both heads during engagement and disengagement, as without NIPBL we would expect them to separate to more than 10 nm. The similarity of the distance to which the heads separate with and without NIPBL at high forces (>20 pN) suggests that NIPBL likely does dissociate from at least one of the heads at high forces, but possibly stays connected with both heads at lower force.

c. If the NIPBL spring were to store the 18 kcal/mol energy provided by head dimerization, and the power stroke supposedly delivers 22 kcal/mol (in the presence of NIPBL), the total energy expenditure in the experiments would be 22+18 kcal/mol (displacement against the bead force and the NIPBL spring), which substantially exceeds the hydrolysis energy provided by 2 ATP molecules, <30 kcal/mol (lines 303-310). The parameters presented do not add up to a thermodynamically possible course of events.

The proposed mechanical coupling of head-head engagement with compression of a NIPBL-loaded spring are thus not supported by the physical parameters presented by the authors and result at least to a very significant extent from ignoring the bound DNA.

We thank the reviewer for these considerations. In the revised manuscript we have performed additional simulations, which showed that in the presence of DNA NIPBL spring can store less energy than we originally predicted in the absence of the DNA. Our new simulations reveal that in the presence of DNA, bent NIPBL plus DNA store ~ 6 kcal/mol more energy than relaxed NIPBL and DNA (New Figure 3). This number is substantially smaller than 18 kcal/mol predicted by our simulations without DNA, which is due to stronger interaction between NIPBL and DNA in the bent form than in the relaxed form. Thus 22 + 6 kcal/mol is much closer to the 30 kcal/mol for the energy of hydrolysis of 2 ATP molecules.

We also performed additional experiments in the absence of DNA for which the energy of NIPBL "bending" is expected to be ~ 3 times higher based on our simulations. Interestingly, efficiency of the head engagement against the external force without DNA was strongly reduced (New Fig. 2e). This is consistent with our prediction that in this case all energy available from ATP binding would be used on NIPBL bending and thus, no free energy would be left to counteract the action of the optical tweezers. Thus, without DNA the energy required to bend NIPBL is likely too large to be overcome in addition to the external force generated by the optical trap that opposes the engagement. However, in the presence of DNA, force required to bend NIPBL becomes smaller, and the available free energy can now be used to generate head engagement against external forces applied by optical tweezers.

Specific comments:

1. In Fig. 1C-D, the authors only show data for a single condition (+NIPBL+ATP+DNA) for the conformational changes of the SMC coiled coils. It is unclear whether these changes are dependent on ATP or DNA. The authors should compare the conformational dynamics without DNA or without ATP, as well as in a locked ATP-bound state (such as a hydrolysis-deficient mutant or using non-hydrolysable nucleotide analogues). Rate constants for each state transition should be determined and presented. Whenever conditions were tested, the authors need to provide the data. Even from

non-dynamic molecules, extension and relaxation experiments can still provide meaningful information.

We thank reviewer for these suggestions, which we followed. We performed additional control experiments excluding NIPBL, DNA or ATP. We now show that at the 1 pN force, discrete transitions between three head-hinge states separated by 15 and 17 nm require all three ATP, DNA and NIPBL (New Figure 1e and S2a). We also found that the head-hinge movement can occur in the presence of NIPBL, DNA and a non-hydrolysable ATP analogue AMPPNP at 1pN. We conclude that in the ATP bound state and in the presence of NIPBL and DNA the coiled-coils adopt relatively stable conformations which can be clearly resolved at 1pN. Similar to the ATP-bound state, forces as low as 1.5 pN are sufficient to fully extend cohesin in the unbent state in the presence of AMPPNP. These findings support our original interpretation that transitions between three mechanical head-hinge states are driven by Brownian fluctuations and not power by chemical energy of ATP. Corresponding corrections were incorporated in the revised text and new figures demonstrating results of the control experiments have been added to the Supplementary material. The rate constants for transitions determined by fitting to the model are now presented in Figure legends.

We also thank the reviewer for suggestion to analyse non-dynamic molecules. We did not include static tethers into the analysis of cohesin conformations because the absolute resolution of our experiments does not allow measuring the length of cohesin with necessary precision to determine its state (bent or unbent, engaged or disengaged) when there is no transitions. We now show in the revised version. To illustrate this, we attempted measuring the absolute length of the head-hinge cohesin at 1.5 pN force as we know from our experiments on Figure 1 that at this force the molecule is fully extended and doesn't undergo any transitions. Thus, we expected to measure the distance of ~ 50 nm. However, our absolute measurements indicated large variability in the determined lengths of the cohesin from 12 all the way to 65 nm (New supplementary Fig. 6g). This is because measurements of the absolute length depend on the exact values of the bead radius and the bead height, which have uncertainties associated with them, which we now quantified in the updated version of the manuscript. These uncertainties strongly affect absolute measurement, but cancel each one out and almost do not affect the relative measurement (New supplementary Fig. S6f,g). This shows that the precision with which we could determine the absolute length of cohesin would not be sufficient to distinguish the length of the bent versus extended molecules that do not undergo dynamic changes. On the other hand, the relative change of the tether length can be determined with nanometer precision even at forces as low as 1pN which allows to resolve different states for molecules that dynamically switch conformations. We included examples of dynamic and static tethers in the supplementary material for the revised version (New supplementary Fig. S2a and S3b).

2. For the SCC1 TEV cleavage experiments in Fig. 1E, the authors need to specify where is the TEV site is located. The integrity of the cleaved complexes should be demonstrated (for example by size-exclusion chromatography and SDS-PAGE to demonstrate that these complexes still contain all components, including NIPBL). The proposed model can be more specifically addressed by lengthening or shortening SCC1 and demonstrating a corresponding change in the intermediate state, or by identifying and mutating the SCC1 residues involved in the proposed contacts.

We thank reviewer for these suggestions, which we followed. In the revised manuscript we have added the detailed information about the 3xTEV site, which was positioned at 471 AA residue of SCC1 analogous to the previously published construct [Davidson et al., 2016 EMBOJ].

Indeed, the compromised integrity of the TEV cleaved cohesin is a possibility which we did not consider in the original manuscript, but we agree with reviewer that it can explain our data and together with other published data represent a scenario, which we now consider likely.

In the revised manuscript we confirmed that TEV cleavage indeed took place in our conditions. To do this, we loaded the same cohesin as we used for the force measurements on lambda phage DNA tethered via ends on the surface of the coverslip and we verified that most cohesin molecules remained on DNA after high salt wash (600mM NaCl). We then performed the TEV cleavage reaction under conditions identical to TEV cleavage of the surface-tethered cohesin and introduced 600 mM NaCl again. As was shown previously [Davidson et al., 2016] we expected that cohesin that was cleaved should be removed from DNA. Indeed >85% of cohesin was removed after the high salt wash confirming the successful TEV cleavage. Thus, we are confident that TEV cleavage does occur in our conditions and deactivation of the head-hinge movement in our assay is indeed likely due to the TEV cleavage (New Supplementary Fig. 2b).

However, taken all observations together, we have now shown that the TEV cleavage in cohesin does both, it inactivates the head-hinge movement (New supplementary Fig. 1e) and disrupts the salt-resistant cohesin-DNA interaction (New supplementary Fig. 2b). Moreover, TEV cleavage of a similar human cohesin construct is also known to abrogate its DNA loop extrusion activity [Davidson et al., 2019, Science]. Thus, we agree with the reviewer that a likely interpretation is that the integrity of this cohesin construct is indeed possibly compromised after the TEV cleavage, which explains the lack of any of cohesin's activities by the cleaved complex. We have made this clear in the revised version of the manuscript.

3. The schematics in Fig. 1F should make it explicit that the experiment provides no information on the relative position of the second, non-trapped (SMC1) head. This should be indicated in the text (line 160). Bending of the coiled coils are observed in cryo-EM structures at both the joint and elbow, which should at least be outlined in the text.

We thank the reviewer for this suggestion which we followed. We have shaded SMC1 in the Figure 1 and better explained in the figure legend that the position of the SMC1 head is unknown in this experiment.

4. The experiments in Fig. 2B lack a comparison of +NIPBL-ATP and -NIPBL -ATP conditions, as well as the dependence on DNA. These data need to be included.

We thank the reviewer for the suggestion, which we followed. First, we performed experiments without NIPBL. As we now show in the revised manuscript, in the absence of NIPBL we did not observe any spontaneous 10 nm steps even in the presence of ATP. Thus, NIPBL is required for the head-head engagement/disengagement consistent with Bauer et al., 2021. In addition, we show that in our mechanical experiments the head-head linkage without NIPBL appeared flexible and stretched with increasingly applied force (Figure 2b). When NIPBL was added, the head-head linkage became much stiffer, indicating that NIPBL increases the stiffness of the mechanical compliance between ATPase heads. We also performed additional experiments in the presence of NIPBL and absence of ATP and now show that in this case we occasionally detected a single disengagement step, but never observed engagement against external force (New Figures 2e, S3b,c). Thus, we show that NIPBL and ATP are both required for the heads to engage against external force. These data are consistent with earlier smFRET data showing that both NIPBL and ATP are required for the head-head transitions (Figure 3B in Bauer et al., 2021).

5. The disengagement rate plotted in Fig. 2E is a combination of disengagement of the SMC1:2ATP:SMC3 module (k_{off}), the rate of ATP hydrolysis, and the rates of Pi and ADP release. Some of these aspects can be dissected by repeating the experiments with mutant cohesin complexes incapable of hydrolysing ATP.

We thank the reviewer for this suggestion. We have performed additional experiments in the presence of non-hydrolysable ATP analogue AMPPNP, and we could not detect any dynamic molecules (New Fig. 2e). The likely explanation of this is that after the addition of AMPPNP majority of the molecules goes relatively quickly into the engaged state and do not disengage. However, since we can only measure one molecule at a time, by the time we perform measurement even with the first molecule, there is no more dynamics left to be observed by our method and therefore all traces we see are static.

As reviewer suggested earlier, the state of the molecule (engaged or disengaged) could potentially be inferred from the overall length of the tether for static traces. However, as we now show in the revised version the error of determining the absolute length of the tether is much higher than the error of measuring the relative distance change in dynamics traces. This is because as we discussed in the response to a different point above, the absolute resolution depends on the absolute size of the bead and the absolute distance between the bead and the coverslip (New Supplementary Fig. 6c,f). Our new measurements also show that the resulting error in determining the absolute length of cohesin is significantly larger than the size of the step that we expect and therefore, we are unable to determine the state of the molecule from the static trace (New Supplementary Fig. 6g). Given this limitation and these new data, we reasoned that we would not be able to distinguish between potential states of the mutant cohesin complexes incapable of hydrolysing ATP, which are expected to be largely static.

6. The SMC head-head interaction is a simple binding event, where the binding probability (and k_{on}) at constant ATP concentrations depend on the collision of the two heads, which in the setup depends directly on the movements of the beads attached to one of them ($\langle x^2 \rangle = kT / \kappa$) and keeping them an average of 10 nm apart, and thus scales with the trap stiffness $\sqrt{kT / \kappa}$, rather than the force exerted, $e^{(\delta F / kT)}$. Because the authors change the trap stiffness whenever the pulling force is altered, the engagement rate gradually drops (Fig. 2F). This can be easily demonstrated by repeating the experiments in Fig. 2F at the same trap stiffness.

We apologise for the confusion in describing our methods. We did not change the trap stiffness during our experiments, which we now clearly explained in the revised methods section. The formula cited by reviewer describes the variance of the bead fluctuation driven by thermal motion ($\langle x^2 \rangle = kT / \kappa$). The graph on Fig. 2f does not show the variance of coordinate, but the rate between individual 10 nm stepping events shown on Fig. 2c, which is proportional to the inverse time between the individual engagement and disengagement events now indicated on Fig. 2c for the 5 pN trace. To illustrate that the steps that we see are not related to the trap stiffness and Brownian noise on the figure 2 we have plotted the step size as the function of force and also separated the engagement and disengagement steps. One can see that there is no correlation between the measured steps size and externally applied force and the step size is similar for engaging and disengaging events.

Figure 4. Left - Dependence of the step size measured for the head-head construct as the function of the applied force (steps sizes for force >15 pN are for disengagement events only). Right - Engagement and disengagement size steps plotted separately.

7. Along these lines, removal of NIPBL and its contribution to restraining the extension of the heads at 15 pN, results in an average spacing of the heads 20 nm apart, which becomes hard to overcome by the bead fluctuations.

We agree with the reviewer that NIPBL acts as an additional mechanical scaffold between ATPase heads, which restricts and coordinates their movement. The reviewer is also correct that the resolution of our experiments is limited by the Brownian noise of the bead. However, the resolution is also improved with the increased force because mechanical tension reduces Brownian fluctuations. At low force and in the case of the head-hinge movement, our resolution was typically in the range of 5 nm as measured by the standard deviation (New Supplementary Fig. 1b), which is still better than 20 nm cited by the reviewer, and it allowed us to determine 15 and 17 nm steps in the head-hinge movement. However, the experiments referred by the reviewer were done at even higher forces of 10 and 20 pN at which the resolution was better than 2.5 nm (standard deviation of the signal) as we quantified in a new Supplementary Fig. 1b. Thus, 20 nm movement referred by the reviewer is way above our resolution and can be easily detected above the Brownian noise. Figure 2b shows that the head-head linkage without NIPBL behaves as a stretchable chain as would be expected for more-or-less any biological molecules under mechanical tension. Higher applied force leads to longer extension between heads. It also shows that in the presence of NIPBL the stiffness increases, and the head-head linkage becomes essentially non-stretchable at these forces. Both experiments on Figure 2b were done at the same trap stiffness. We have made this clear in the revised version of the manuscript.

8. The authors should assess the dependence of the head engagement rate on ATP concentrations to discern whether ATP molecules need to bind simultaneously or sequentially.

We thank the reviewer for this suggestion. Indeed, a single-exponential dependence of the head engagement rate on ATP concentration may indicate that only one ATP binding event is required for the engagement, whereas higher order dependence would suggest that both ATP molecules could be required. Since we can only perform experiments with one molecule at a time, we first investigated how many successful molecules must be recorded in these experiments to make the statistical argument and either accept or reject the single-exponential dependence.

In our experiments, successful recordings at 1mM ATP yielded an average time between engagement steps of 0.66 inverse seconds (corresponding to an average interval of 1.5 s between engagement events) with a standard deviation of 0.52 s^{-1} and a standard error of the mean 0.14 s^{-1} . We did not observe any transitions in the absence of ATP, so we assumed at 0 mM ATP engagement times are above our observation length, which was typically 200 s or above. Using these data, we

simulated times between engagement events (reverse engagement rates) for 5 different ATP concentrations in the 1000-fold range from 1 μ M to 1 mM (See below - Figure 3). Number at the top right of each graph shows the number of individual engagement events used for the simulation for each ATP concentration. For example, when only 5 events are simulated for each ATP concentration the resulting error bars between multiple engagement times are high (shown as standard errors of the mean) and the fitting (solid line) is poor, which does not allow to accept or reject the single-exponential dependence. As number of trials increased, the error bars decrease. The solid line is the single-exponential fit of the data. Figure 3 (below) shows that the number of trials required to determine whether data can indeed be fit with a single-exponential is in the N=500 range. This means that in order to tell whether ATP dependence indeed follows a single-exponential behaviour and exclude possible higher-order dependence we would have to collect \sim 500 engagement events at 5 different ATP concentrations. Many of these engagement events would also be at low ATP concentrations suggesting much longer waiting times between them. From the data on the Figure 2, the total recording time of 500 events at 50 concentrations would be \sim 65 hours. This recording time must only come from successful dynamic beads, which typically represent less than 20% of all our recordings (Figure 1e and 2e).

This required number of recordings is \sim 130 times of what we have managed to collect for the 1 mM ATP head-head experiments presented in the paper, which took us over 7 months of continuous experiments. Thus, unfortunately, these calculations show that due to slow engagement dynamics and low throughput nature of these single-molecule experiments it would be technically impossible for us to collect sufficient statistics at different ATP concentrations to be able to distinguish whether the data could be fit by a single-exponential.

Figure 3. Simulated engagement times as function of ATP concentrations. The number in the upper top of each graph indicates the number of simulated engagement events for each ATP concentration. Solid lines are single-exponential fits.

9. To demonstrate that the 10-nm step indeed measures ATPase head disengagement, the authors could make use of a recent construct that allows locking the ATPase heads together by addition of rapamycin (cohesin SMC1-FKBP/SMC3-FRB; Bauer et al., 2021).

We thank the reviewer for this suggestion. Indeed, we would expect that addition of the rapamycin to this complex would be expected to block the head-head transitions. However, since we can only look at one molecule at a time and the addition of the rapamycin would lock all molecules in the flow cells and make them static, and as we discussed in a response to a different point above, our technique does not allow us determining the state of the static molecules. This is also biochemically challenging, because in addition to the FRB and FKBP tags we would have to have two additional tags in the same place that we would use for the attachment of cohesin to the surface on one side and to the optically trapped bead on the other.

However, our other experiments do show that 10 nm step is the step involved in head engagement. We find that similar to Bauer 2021, both NIPBL and ATP are absolutely required for the head-

engagement as detected by the optical trap in our experiments (New Figure 2e). We also show that 10 nm engagement does not occur in the absence of either NIPBL or ATP. We find that occasional disengagement events are detected in the absence of ATP, which is never followed by the engagement (New Supplementary Fig. 3b,c). Therefore, dynamic transitions between heads observed in our experiments very likely correspond to the engagement movement of ATPase heads.

10. The structure models depicted in Fig. 3A-B need to show DNA whenever there is actually DNA present in the structures. I also recommend including a DNA molecule in the molecular dynamics calculations to determine its contribution to stabilising the ‘bent’ state.

We thank the reviewer for this suggestion, which we followed. We have now performed two types of simulations with and without DNA and clearly indicated the position of the DNA molecule in simulations (Revised Figure 3). We find that in both cases the energetics is qualitatively similar, and the bending of NIPBL is associated with the increase of the free energy. However, we also find that in the presence of DNA the free energy change associated with the NIPBL bending is significantly smaller. This is presumably because in the “bent” state there is a stronger contact area between NIPBL and DNA, which results in tighter binding contributing to the overall lower energy in the “bent” state.

11. The coiled-coil handle loop extrusion experiment shown in Fig. S3G lacks statistics for the observed frequency of loop extrusion as well as controls that ensure the observed activity can be attributed to the labelled population. Either the labelled protein should be separated from unlabelled (e.g., by size-exclusion chromatography), or a fluorophore should be included in the handle and co-imaged with the DNA.

We thank reviewer for the suggestions, which we followed. In the revised manuscript, we have included statistics of the loop extrusion observation frequency in the updated Methods section. As suggested by the reviewer, we now also performed control experiments, in which we labelled cohesin at the coiled coil by attaching a quantum dot at the coiled-coil end opposite to cohesin (New Supplementary Fig. 7d). This way only the quantum dot labelled cohesin attached to the coiled coil is visible in the loop extrusion assay. Indeed, we observed that quantum dots localised at the stems of DNA loops, confirming that the handle-coupled cohesin retains DNA loop extrusion activity. The section regarding the additional experiment was added to the Methods.

12. The references to the Lee et al., 2020 and Collier et al., in lines 73 and 75 seem to be swapped.

We thank the reviewer for pointing this out. Indeed, Lee et al., 2020 was incorrectly cited on line 73, which we replaced with Collier et al.

13. Contrary to what the authors state (lines 75-80), the smFRET data from Bauer et al., 2021 shows that the hinge-head distance is an ensemble irrespective of the presence NIPBL or ATP. Moreover, cryo-EM structures of cohesin in the absence of ATP also show the hinge in the vicinity of the SMC3 head (Petela et al., 2021). This section is a misrepresentation of the available structural evidence.

We apologize for the confusion, which at least partially stems from the disagreement that currently exists in the literature. We agree with the reviewer that Cryo-EM data unambiguously show the hinge in the vicinity of the SMC3 head in all available structures. However, Figures 6A and 6G from Bauer et al., show that the hinge may detach from ATPase heads in the presence of ATP and NIPBL. On page 11, authors speculate that: “hinge bending and head engagement occur in a mutually

exclusive manner”, which contrasts with Cryo-EM structures. We highlighted this contradiction better in the introduction.

14. It should be possible to establish at which distance the static state is fixed in the absence of NIPBL from force-extension measurements and this data should be included (lines 121-122).

We thank the reviewer for the suggestion. In the revised manuscript, we have added example data of the head-hinge distance measured without NIPBL (Supplementary Fig. 2a). These traces do not show dynamic transitions, and as the reviewer suggests their state (bent, or unbent) would have to be determined from their absolute length. However, as we discussed in response to a different comment above, our additional analysis showed that although we can measure relative changes in the distance with very high precision, our resolution is much more limited for measuring the absolute length. (New Supplementary Fig. 6g). Therefore, we could not determine the absolute head-hinge distance with the precision required to unambiguously assign a bent or unbent state to static traces.

Reviewer #2 (Remarks to the Author):

Cohesin is a member of the SMC family of proteins and plays crucial roles in organization of the 3D genome in cellular functions. In this manuscript, Pobegalov et al. used optical trapping, combined with computational methods, to investigate conformational dynamics of cohesin.

Overall, the authors presented solid evidence for the 3-state, thermally driven, coiled-coil bending of the cohesin arm. Arguments based on the observed 10 nm engagement/disengagement step of the ATPase heads and its energetic implications, as shown in molecular dynamics simulations of relaxing a bent NIPBL subunit, are compelling. Based on these findings, the authors proposed a model in which energy from the engagement of ATPase heads is stored in the bent NIPBL subunit and its release allows the protein to overcome the activation barrier in local DNA bending when initiating loop extrusion. The text is clearly written and flows logically throughout.

We thank reviewer for positive assessment of our work. In the revised text we have addressed all comments and added the suggested controls, which we believe significantly improved our manuscript.

The following, many minor, points should be addressed/clarified regarding some specific statements in the text.

1. In Fig. 1D, the peak centers of histogram at 0.5 pN are separated by a distance of >20 nm between and completely folded and intermediate states. Can the authors clarify this discrepancy compared to the stated 17 nm step (as marked by grey dashed lines)?

Indeed, this point was not sufficiently clear from our original manuscript, which we corrected in the new version. The reason for this discrepancy is the combination of the lower resolution at lower stretching forces and the overall shorter length of the extended cohesin at lower force. The resolution and therefore our ability to distinguish steps is different at different forces because it is limited by the Brownian noise of the bead and the flexibility of the SMC coiled-coils. Stronger forces stretch SMCs more, which results in smaller fluctuation of the bead and higher resolution, which we now quantified in the new Supplementary Fig. 1b. In addition, the length of the whole coiled coil

handle plus the head-hinge cohesin in the fully extended state also depends on the external force as indeed is expected for a flexible molecule and as we now show in the New supplementary Fig. 1f. At 1 pN the system is already almost fully stretched and increasing the force to 1.5 pN does not extend the whole complex further. However, at 0.5 pN unbent cohesin plus linkers in the system make it ~ 10% shorter (New Supplementary Fig. 1f). Thus at 0.5 pN force, the original peak at ~ 20 nm is the result of the intermediate and small fraction of fully extended states that cannot be resolved from each other because they are closer to one another due incomplete extension of the molecule at this force and because the noise at low forces is higher. We have added this information to the revised manuscript.

2. In discussing the experiment involving TEV-cleavage of SCC1 leading to a static cohesin molecule under 1 pN force (Fig. 1E), the state is assumed to be fully extended (Line 174). Can the authors explain how the particular state is assigned over the other two? As the authors state in the Methods section (Lines 601-603), traces that were static would have the uncertainty in state assignment?

This point was indeed not sufficiently explained in our original manuscript. Reviewer is correct that the state of static traces cannot be assigned based solely on our data only due to uncertainty in determining the absolute length of the tether. Our original interpretation was to assume that TEV cleavage removed the limitation restricting SMC bending and therefore at 1 pN all TEV cleaved cohesins must be unbent.

However, we agree with the reviewer that this is only one possibility and TEV cleaved cohesin may be locked in a bent or intermediate state. Our additional experiments showed that the TEV cleavage in cohesin does both - it inactivates the head-hinge movement (Fig. 1e, S2a) and it disrupts the salt-resistant cohesin-DNA interaction (Fig. S2b). Moreover, TEV cleavage of a similar cohesin construct is also known to abrogate its DNA loop extrusion activity [Davidson et al., 2019]. Thus, it appears that TEV cleavage makes cohesin inactive in every assay. We highlighted this in the revised version of the manuscript and explained that the state of the non-active TEV cleaved cohesin is unknown.

3. Controls:

a. Regarding the experiment shown in Fig. 1E, TEV cleavage is assumed to have taken place, which in turn resulted in a change in cohesin behavior under force. While not unreasonable, it would be more convincing to show that under the experimental condition used (introducing TEV after immobilizing cohesin), TEV did indeed cleave and not just merely affect the force behavior (i.e. make cohesin arms static) some other way.

We thank the reviewer for this suggestion, which we have followed. To show that the TEV cleavage indeed took place, in the revised manuscript we used the same experimental conditions in a modified assay in which it was easier to monitor the efficiency of the cleavage. In this experiment (see new Supplementary Fig. S2b) we loaded the same cohesin as we used for the force measurements on lambda phage DNA tethered via ends to the surface of the coverslip using the conditions promoting topological loading and we verified that most cohesin molecules remained on DNA after high salt wash (600mM NaCl). We then remove high salt, performed the TEV cleavage reaction under conditions identical to TEV cleavage of the surface-tethered cohesin and reintroduced 600 mM NaCl again. As was shown previously we expected that cohesin that was cleaved should be removed from DNA [Davidson et al., 2016]. Indeed >80% of cohesin was removed after the high salt wash confirming the successful TEV cleavage. We added this data to the revised manuscript.

b. In Supp Fig 3G, the authors show that CC-handle coupled cohesin is active in a DNA looping assay using surface tethered DNA and cohesin in solution. Considering that the configuration of optical trapping experiments used surface-tethered cohesin, it would be better to show activity with that particular attachment geometry, or at least explain in the Methods why the control is carried out somewhat differently.

Reviewer is correct that the geometry of these two experiments is different. When we applied force to cohesin, it had to be tethered to the surface so that we could stretch it. In the loop extrusion assay DNA is typically tethered to the surface and cohesin is free to move on DNA and extrude loops. It was not technically possible for us to see both activities in the same assay for the following reasons. If loop extrusion assay could be done with cohesins tethered to the surface, then DNA would have to be free – Both DNA and cohesin cannot be tethered at the same time as that would not allow for the relative movement between them. However unbound DNA has persistence length of 50 nm and therefore it coils on itself and adopts highly folded conformation. In loop extrusion assays, lambda DNA is normally stretched to 11 – 14 microns on the surface to allow for the loop visualization. However, the radius of gyration for lambda DNA is 0.9 microns, which means when it is not stretched, it forms a blob of approximately 0.9 microns in size. If loops were to form in such highly intertwined DNA, it would not be impossible to resolve them with conventional microscopy. Therefore, we opted for different geometries in these two experiments. Importantly we did show that our cohesin is active in both assays – it extrudes loops, and it undergoes conformational changes that generate force. We assumed that the activity of cohesin must remain the same when it is active regardless of the assay used. We agree with the reviewer that it would have been much better to see both activities in the same assay. However, this wasn't yet technically possible.

4. In Line 143, the authors stated that the angle formed at the hinge in the intermediate state is ~60 deg. It would be helpful to briefly explain in the Methods how that is calculated.

We thank the reviewer for pointing this out. We originally calculated the angle of 60 degrees by assuming that SMC coiled-coils can be represented by two stiff segments connected by the flexible elbow. We took the lengths of the SMC segments from available Cryo-EM data and the angle was calculated from all known sides of the triangle.

However, this representation is an oversimplification, and it doesn't take into account other flexible parts in coiled-coils. In the revised version we decided to remove this angle and instead focus on the distance that we determine experimentally, which is the distance between the head and the hinge. Position of the rest of the molecule as well as the shape of the SMCs is unknown in our experiments, which we now clearly indicated.

5. Compared to head-hinge movement measurements, head-head engagement dynamic experiments were carried out in the absence of DNA. Can the authors comment on what specifically led to the decision of not including DNA in those experiments?

We thank the reviewer for pointing this issue, which was not sufficiently explained in our original manuscript. Like the head-hinge experiments, head-head experiments were originally also performed in the presence of DNA. We have corrected our text and updated Methods section to explain this clearer. In the revised manuscript we have now also done additional experiments in the absence of DNA for both head-hinge and head-head complexes. The comparison between these experiments performed in the presence and absence of DNA is now clearly presented in Figures 1e and 2e.

6. In Fig. 2B, under the +NIPBL condition, did the author observe a rip of ~10 nm in the force ramp up to 20 pN? Not sure why the force-extension plot is omitted for force < 10 pN. It could be more interesting to show an example of a rip/head disengagement in this plot, if indeed it did happen on a regular basis, in the presence of ATP and NIPBL.

The reviewer is correct that in some cases we could indeed observe 10 nm steps while producing force-extension curves. We have explained this in the revised version of the manuscript, and we have now included an example trace to illustrate this (new Fig. S3a). Since in the presence of NIPBL the head-head linkage appeared very stiff (revised Figure 2b) forces below 10 pN resulted in the change of the distance, which was below the resolution of our instrument. Therefore force-extension curves were recorded at forces above 10 pN. In the revised manuscript we recorded force-extension at lower forces, which is now presented in the new Supplementary Fig. 1f.

7. Regarding experiments presented in Fig. 2C, what percentage of molecules showed 10 nm transitions at forces <20 pN in the presence of NIPBL and ATP? (Line 213-215). In addition, it could be helpful to show an example trace at 15 pN in 2C, in addition to 5 pN and 10 pN, since it is stated that the transitions occur up to 15 pN (Lines 221-222).

We thank the reviewer for the suggestion. We have added the new data suggested by the reviewer to the revised manuscript (Fig. 2c) and the figure legend for Fig. 2e.

8. Regarding Fig. 2D, can the authors comment on if there was any correlation between force load and observed step size? Or correlation between an engagement distance vs. the following disengagement distance (or the other way around)?

We followed the reviewer's suggestion and analysed our data for the proposed correlations. Our analysis did not reveal significant correlation neither between the size of the step and the applied force nor for engagement/disengagement events. The data below shows the step size as a function of force for disengagement events and a separate graph showing all forces combined but split into engagement/disengagement events. As one can see, the variance of the data is quite high for each condition.

Figure 4. Left - Dependence of the step size measured for the head-head construct as the function of the applied force (step sizes for force >15 pN are for disengagement events only). Right - Engagement and disengagement step sizes plotted separately.

9. The authors stated that in the absence of ATP, there was no observed engagement transitions under force (Lines 251-252). Have the authors tried varying ATP concentrations to see if rates of engagement or disengagement depended on ATP concentration? Similarly, have the authors tried nonhydrolyzable ATP analogues? Would the authors expect it to affect disengagement rate? Would

blocking hydrolysis strengthen the argument that the head engagement itself is not dependent on hydrolysis and/or that disengagement is post-hydrolysis?

We thank reviewer for these suggestions, which we followed. We first prioritized collecting more data for the 1 mM ATP and no ATP experiments. Our additional experiments in the absence of ATP confirmed the absence of dynamics traces, but also revealed that in the small number of traces (4 out of 24) contained one disengagement event, which was never followed by an engagement (new Supplementary Fig. S3b,c). Since our buffer did not have any special system for depleting ATP, those molecules may have represented cohesins which originally had ATP associated with them. This suggests that these molecules likely undergo disengagement, but because there is no free ATP their ATPase heads cannot engage. Thus, presence of ATP would primarily affect engagement, but not disengagement rate. This is consistent with the earlier experiment suggesting that engagement is promoted by ATP binding.

We have also followed the reviewer's suggestion and performed experiments in the presence of nonhydrolyzable ATP analogue AMPPNP. Our results from the head-hinge construct shows dynamics in the presence of AMPPNP, while the head-head construct does not (New Fig. 1e and 2e). However, as we now discuss in the revised manuscript, we could not assign the state to static traces. In other words, if the trace was static, we could not tell whether the heads were engaged or disengaged for the head-head cohesin, or whether the head-hinge cohesin was bent or extended. This is because as we now show, the error of determining the absolute length of the tether is much higher than the error of measuring the relative distance change in dynamics traces, since the absolute resolution depends on the absolute size of the bead and the absolute distance between the bead and the coverslip. To illustrate this, we attempted to measure the absolute length of the head-hinge cohesin at 1.5 pN force as we know from our experiments on Figure 1 that at this force the molecule is fully extended and doesn't undergo any transitions. Thus, from the structural studies we expected to measure the distance of ~ 50 nm. However, our absolute measurements indicated large variability in the determined lengths of the cohesin from 12 all the way to 65 nm (New Supplementary Fig. S6g). Thus, our new measurements show that the resulting error in determining the absolute length of cohesin is significantly larger than the size of step that we expect and therefore, we are unable to determine the state of the molecule from the static trace.

10. Line 598 in the Methods section refers to "Supplementary Figure E, F". Presumably it should be "Supplementary Figure 3 E, F".

We thank reviewer for bringing our attention to this typo, which we corrected.

11. Legend for Fig. 1C (Line 927), 1D (Line 931), 1E (Line 933), and Fig. 2B (Line 940). These corresponding figure panels have axes labeled "relative distance/extension". The text in figure legend does not specify "relative".

We thank the reviewer for pointing this out this error, which we corrected in the revised version.

12. Legend for Fig. 3B, Line 954, "Bend" should be "Bent". In the following sentence, the red appears to represent "centre of mass distance" rather than 'centre of mass' as stated?

We thank the reviewer for pointing this out. We corrected this in the revised manuscript.

Reviewer #3 (Remarks to the Author):

Pobegalov et al. use optical tweezers to examine the energetics and dynamics of cohesin complexes in a surface-tethered assay. They observe bending of the cohesin hinge with an intermediate state and describe its force dependency with a 3 state model. Furthermore, they use a similar assay to monitor cohesin-head-head dynamics, which they describe as a power stroke able to generate up to 15 pN in force. Using MD simulations, they find that a part of this mechanical energy is stored in the conformation of NIPBL-Scp2. They discuss these results in the context of the mechanism of loop extrusion with the further support by MC simulations. They speculate that the two different molecular movements they observed in their experiments play distinct roles in loop initiation and loop elongation.

The manuscript is very exciting. It provides clear mechanistic insight into this very relevant topic. The research is designed well and conducted carefully, the findings are very interesting and well interpreted, and the paper is written very well. The findings will be of interest to the readers of this journal.

We thank the reviewer for the positive assessment of our work.

I have a few minor remarks/questions:

- It would be helpful for the assessment of the data if the standard deviation of the force during force clamps would be given or sample traces would be shown in the SI.

We thank the reviewer for this suggestion, which we followed. We now show example traces that include measured forces at different set force (New Supplementary Fig. S1b).

- Fit parameters for Suppl Fig 1F are not given, and would be potentially interesting

We thank the reviewer for pointing this out. We have now included the inferred parameters to the corresponding legend of the new Supplementary Fig. S2d,e.

- The data in Fig 2B is described in the text as elastic with a constant stiffness of 0.7 pN/nm, however, the data and the not further specified trend line both look rather non-linear.

We apologize for this confusion, which we corrected in the revised manuscript. Indeed, the trend is expected to be non-linear because the extension of the molecule under the range of external forces should follow a worm-like chain behaviour. The 0.7 pN/nm stiffness we originally indicated in our text was referred to a linear approximation. We agree with the reviewer that this number was confusing. We removed it from the new version of the text and replaced the the worm-like chain fit, which parameters are now indicated in the legend.

- The exponential fit in 2F with a fixed length change is slightly misleading since commonly this parameter is found to be smaller than the associated structural change. Also, the (linear?) trendline is not very helpful as an exponential dependency is expected. Maybe a better approach would be to perform an exponential fit with the length change as a fit parameter. I would expect the best fit length change to be much smaller than 10 nm, which could then be discussed. I nevertheless agree with the authors that the weak force dependency and the head engagement happening even at 15 pN favors a power stroke mechanism over a Brownian ratchet mechanism.

We thank the reviewer for these suggestions, which we followed. We have replaced trendlines in the new Fig 2f,g with exponential fits. As the reviewer expected these resulted in $\Delta \sim 0.1$ nm, which

was inconsistent with the measured 10 nm step. We have discussed this inconsistency in the text as reviewer suggested.

- The authors compare the energy they estimate for the power stroke with the elastic bending energy of NIPBL-Scc2. However, they measured the power stroke energy in the presence of NIPBL-Scc2. I think this needs to be considered, since the power stroke needs to overcome both the force applied by the optical trap as well as the bending energy of NIPBL-Scc2. This can also be seen in figure 4B, where the lower line should presumably read $\Delta G = \Delta G (\text{ATP binding}) - \Delta G (\text{Scc2 bending})$.

The reviewer is correct that we are measuring the power stroke energy in the presence of NIPBL^{Scc2}. Thus, the amount of energy expended during the engagement should be the sum of energies required to bend the NIPBL^{Scc2} and the force applied to the bead in the optical trap multiplied by the distance the bead travels. We have stated this explicitly in the revised manuscript.

- Page 8 "Thus, our data combined with previous observations suggest that the energy that leads to the force generation during the head engagement arises from ATP binding, but not hydrolysis." This is not entirely clear to me, the authors could maybe elaborate on this argument.

We have elaborated to make this statement clearer. Briefly, this sentence related to the discussion of the source of the energy that allows ATPase heads to engage against external force and compress the molecule of NIPBL. In the revised manuscript, we discussed that this energy likely comes from ATP and more specifically, it must be released during one of the steps in the ATP cycle. Since ATP binding is the step preceding the engagement, we propose that free energy required for the engagement is released due to ATP binding. As we discuss in the revised manuscript this is consistent with other studies that looked at this experimentally and computationally.

- It is not entirely clear to me where the directionality of loop elongation by the head-hinge-movement would come from, especially if it is entirely driven by thermal motion of the hinge (Fig. 4E). This might go beyond the scope of this manuscript but might be worth discussing.

We thank the reviewer for this suggestion. We agree it is important to discuss this issue and we added our explanation to the revised version of the manuscript. As we now explain in the revised text, the exact mechanism of DNA loop extrusion remains unknown, and although our results reveal how cohesin generates force, they do not allow us to show directly how conformational changes in the cohesin and generation of force are coupled to the cohesin-DNA interaction and movement along DNA.

However, based on our earlier work, we also propose a relatively simple model that can explain how head-hinge movement can drive loop extrusion in one direction. Based on the available structure of the gripping state, we assumed that once the molecule is bent, the hinge is touching the head and forms a strong composite interface which favours DNA binding. From this position the hinge can only go one way – away from the heads. We assume that DNA remains bound to the hinge when cohesin unbends and thus travels away from heads, thus increasing the size of the DNA loop. Because the strong composite DNA binding site is dissolved, DNA is more likely to dissociate from the hinge when it is away from the heads. Once that happens the hinge is free to bend again and bind DNA at the heads. Thus, if the hinge binds DNA mostly at the heads and releases when it is away, movement becomes unidirectional. Our simulations from our previous work reveal that this model indeed provides directionality to the loop extrusion [<https://doi.org/10.7554/eLife.67530>].

- The way the data is analysed is very sound. However, I wonder what the impact of the other geometrical parameters in Equation (1) is. How was the distance of the bead from the flow cell adjusted to 60 nm and what is the uncertainty of this adjustment? Similarly, how was the bead diameter determined and how would a deviation from the assumed bead diameter impact the data?

We thank the reviewer for the suggestion to quantify this more clearly. In the revised manuscript we have added new Supplementary Fig. S6c and S6f, which illustrate the impact of the bead height and bead diameter on the determination of the tether length. The bead position above the surface was adjusted by piezo stage in 10 nm increment steps. The uncertainty in the adjustment came from the uncertainty of determining the absolute z position of the bead, which was done by video tracking while moving the bead in z towards the surface of the flow cell. At z = 0 nm, the bead touches the surface of the coverslip, and further movement results in a change of its image. Supplementary Fig. S6c shows how the images of the bead depend on its axial position, which resulted in the absolute uncertainty in determining the bead z position to ~ 10 nm.

As for the bead diameter, we also assumed that it was 500 ± 30 nm based on the manufacturer specifications. On supplementary fig. S6f we have illustrated how this uncertainty affects the inferred step size. Overall, the effect is small for determining the relative displacements. For example, for the bead height in the range of 50-70 nm the calculated step size is 15.2 ± 0.3 nm and for the bead diameter in the range of 470-530 nm the calculated step size is 15.2 ± 0.5 nm.

Small details:

- Page 7, line 330: comma after however is missing
- Page 2, line 117: If the authors could give concentrations here, it would be rather helpful to the reader.

We thank the reviewer, and we added this information to the revised manuscript.

Reviewer #4 (Remarks to the Author):

The manuscript by Pobegalov et al is a timely and important contribution as it provides measurements critical for understanding SMC loop extrusion dynamics. Importantly, these optical trap measurements under external force are complementary to dynamic AFM and FRET data, providing a wealth of new information and leading to new conclusions. Those include: (i) an existence of an intermediate state; (ii) largely fluctuation driven head-to-hinge dynamics; (iii) generation of a very considerable head-to-head force by ATP binding; (iv) potential storage of the mechanical energy in the deformation of Nipbl molecule. While the manuscript is generally convincing, I have doubts about some conclusions that I challenge below.

In general, I enjoyed reading this very solid and thought-provoking study and I only wish authors were to present more data such as data for different combinations of ATP/DNA/Nipbl (that I'm sure have been generated) and for non-hydrolyzable ATP, that can only strengthen their conclusions.

We thank the reviewer for the positive assessment of our work. In the revised version we have added additional controls requested by the reviewer for both head-hinge and head-head constructs that further strengthened conclusions of the manuscript.

1. Head-to-hinge movement.

Authors imply that the bent and intermediate state at 17nm separation corresponds to the "elbow-

bent states, but observation of the O-looking state for other SMCs in dynamic AFMs may suggest some other conformations. I just suggest authors mention this as a possibility.

We thank the reviewer for this suggestion. Indeed, our experiments allow us to infer the distance between the hinge and the head, but the position of the rest of the molecule as well as the shape of the SMCs is unknown, which we now clearly indicated in the text.

Does the state of the intermediate (i.e. 17nm) depend on the force? It appears that 0.5pN force shows a peak at around 25nm instead. How does this state depend on the presence of ATP/DNA/Nipbl?

Our data does not allow us to accurately measure the size of the step at 0.5 pN. However, we do expect the overall length of the molecule to depend on the external force because under the external load SMCs are expected to behave like worm-like-chains. To illustrate this, we have now included the new supplementary figure 1f, which shows the force extension of the head-hinge cohesin plus other linkages in the low force regime. Our data show that at 1 pN the SMCs together with other linkages are already almost fully stretched and increasing the force to 1.5 pN does not significantly increase its length. However, at 0.5 pN the molecule is notably shorter. As the result, at 0.5 pN force, the original peak at ~ 20 nm (Fig. 1d) is the result of the intermediate and small fraction of fully extended states that cannot be resolved from each other because they are closer to one another due incomplete extension of the molecule at this force and also because the noise at 0.5 pN is higher than at 1 pN, which we now quantified in a new Supplementary Fig. S1b.

Finally, we followed the reviewer's suggestion and included dependence of the dynamic transitions under external force on the presence of ATP/DNA/Nipbl, which is now summarized in new Fig. 1e and example traces are presented in Supplementary Fig. S2a. Briefly, we see that at 1 pN force, all components (ATP/DNA/Nipbl) are required for the dynamic transition between all states.

Authors state that in the absence of ATP/DNA/Nipbl and at high salt the molecule is largely static, but don't show any data, so it's difficult to understand what is this static state (extended? bent?).

In the revised manuscript we have now included examples of the static traces for the both head-hinge and head-head measurements (New Supplementary Fig. S2a and S3b). These traces do not show any transitions and therefore as reviewer points out must remain constantly in one of the states (extended, bent, etc.).

We agree that theoretically it should be possible to determine the state of the static molecule by measuring the absolute length of the complex. However, as we now discuss in the revised manuscript, this was not experimentally possible due to much higher error in measuring the absolute distance comparing to the relative displacement. In other words, if the trace is static, we cannot tell whether the heads are engaged or disengaged for the head-head cohesin, or whether the head-hinge cohesin is bent or extended.

This is because the absolute resolution depends on the absolute size of the bead and the absolute distance between the bead and the coverslip. To illustrate this, we attempted to measure the absolute length of the head-hinge cohesin at 1.5 pN force as we know from our experiments on Figure 1 that at this force the molecule is fully extended and doesn't undergo any transitions. Thus, from the structural studies we expected to measure the distance of ~ 50 nm. However, our absolute measurements indicated large variability in the determined lengths of the cohesin from 12 all the way to 65 nm (Supplementary Fig. S6g). Thus, our new measurements show that the resulting error

in determining the absolute length of cohesin is significantly larger than the size of step that we expect and therefore, we are unable to determine the state of the molecule from the static trace.

What would be the distribution of states at zero force? at physiological 150mM salt?

In fact the model used to fit the data has two adjustable parameters that represent occupancy of the intermediate and bent states at zero force. Reporting these values and comparing with zero force data (if any) could be very instructive.

We thank the reviewer for this comment. We have now included the fit parameters to the legend of the new Supplementary Fig. S2d,e. Our fits show that when interpolated to zero force, at 1 mM ATP, at any time, there are 96% of the molecules in the bent state with only ~4% of the molecules in the intermediate and extended states combined.

We performed all our head-hinge experiments at 50 mM salt because in vitro cohesin is active in loop extrusion assays at this salt concentration and was not shown to extrude DNA loops at 150 mM NaCl or KCl.

I have some doubts in the strength of the main conclusion about entirely thermal motion that drives head-to-hinge dynamics. What the data show is that the system doesn't generate a substantial (>1pN) force on contraction, but I'm not sure one can rule out fast and infrequent power-strokes on expansion. If fast, such movements would not significantly affect statistics of states for the rest of dynamics. Also if dynamics were entirely fluctuation driven it would not get affected by ATP hydrolysis and thus would be similar to dynamics of the system when a non-hydrolyzable ATP were used. It would be great to see data for such experiments.

We thank the reviewer for the suggestion, which we followed. We have investigated the dynamics of the head-hinge transitions in the presence of non-hydrolyzable ATP analogues AMPPNP and surprisingly found that in the presence of AMPPNP, DNA and NIPBL and at 1 pN force cohesin does show head-hinge transitions very similar to those observed in the presence of ATP (Fig. 1e, New Supplementary Fig. 2a). Combined histograms showed that the distribution between bent and unbent states depend on the force similarly as in the presence of ATP (Supplementary Fig. S2d,e) and rate transitions between states inferred from the histograms by fitting to the three-state model, were also similar.

We also found that at 1 pN load there were no discrete transitions in the absence of ATP. Thus, it is likely that the majority of the head-hinge transitions can be explained by thermal fluctuations in the state in which the molecule is bound to ATP.

We agree with the reviewer that our technique may not be able to distinguish an infrequent power-stroke on expansion. If this stroke happens once, for example, after ATP binding, our data show that it will then be followed by multiple transitions between bent and unbent states that happen while ATP remains bound.

2. Head-to-head dynamics

I am concerned about the total work measured ~150pN nm, upon head-to-head engagement as it significantly overshoots the energy of ATP hydrolysis that is ~10kT = 40pN nm.

<http://book.bionumbers.org/how-much-energy-is-released-in-atp-hydrolysis/>

For two ATP that gives 80 pN nm, but still way below 150 pN nm.

Authors suggest that the main energy source is ATP bindings rather than hydrolysis or release. This hypothesis can be more directly tested by (i) varying ATP concentration as it would determine the energy gained by binding; (ii) using non-hydrolyzable ATP or SMC mutants that don't hydrolyze ATP.

Would be great to see the role of DNA in this process. I couldn't find data for +/- DNA, (and not for some other conditions among +/- Nipbl, +/- ATP).

We thank the reviewer for these suggestions, which we followed. First, we performed additional experiments without ATP or NIPBL and confirmed that both molecules are required for the head engagement against the external force (New Fig. 2e). Second, we investigated the effect of the DNA presence and the non-hydrolysable analogue AMPPNP. Our experiments showed no engagement dynamics in the presence of AMPPNP confirming that ATP is the likely source of the energy for the head engagement against external force. Finally, our experiments without DNA revealed that when DNA was not present, the ability of the heads to engage against the external force was also significantly reduced. Thus, the only condition under which heads can engage effectively against external force is + NIPBL + ATP + DNA.

These experiments suggested that the presence of DNA may assist in the formation of the engaged state. Indeed, in the bent state NIPBL makes extensive contacts with DNA backbone which are likely to reduce the overall energy in this state. To further explore this, we have now performed additional simulations of the NIPBL bending with and without ATP and we now show that the bending of NIPBL requires much less energy input when DNA is present (Revised Figure 3). These findings are consistent with our data showing that DNA facilitates the ATPase head engagement and can help engage heads against external force at 5 pN and above.

Regarding the total work measured, we note that our experiments suggest that the energy provided by ATP is spent on two processes during the engagement: NIPBL bending and work against the optical trap during the head engagement. Thus, the total maximal energy that heads must overcome is the sum of the work against the trap $150 \text{ pN}\cdot\text{nm}$ plus energy of NIPBL bending, which is $\sim 6 \text{ kcal/mol}$ ($\sim 42 \text{ pN}\cdot\text{nm}$). Therefore, the total maximal amount of energy that is expected to be released from one ATP in our experiments is $\sim 95 \text{ pN}\cdot\text{nm}$ or 23 kT.

Although this value overshoots the value cited by the reviewer (10kT). However, 10kT seems significantly lower than the reported values of the energy available from ATP hydrolysis. The link provided by the reviewer gives a table of ATP hydrolysis values in different conditions and organisms. The values in Table 1 (rightmost yellow column) indicate the smallest energy of ATP hydrolysis of 14kT and the largest energy of 29kT. Thus, the energy required for the head engagement in our experiments are well within the physiological range of energies and thus can be provided by the ATP hydrolysis.

Storage of energy in Nipbl deformation is plausible, modulus that the total work required can be too large as indicated above. Simulations suggesting the role of head-to-head movement in initiation of extrusion are certainly thought-provoking. I couldn't follow how such a movement can assist in loop extrusion at stages past the initiation -- authors may want to elaborate on this or just explain the initiation is the key step where this movement plays a role.

We apologize this was not explained clearly enough in the original manuscript. The reviewer is correct, our hypothesis was that the energy released during NIPBL decompression could power only beginning stages of the initial loop bending. However, the exact mechanism used by cohesin to extrude loops requires further study - we made this clearer in the revised manuscript.

A very nice paper!
Leonid Mirny

Thank you!

REVIEWER COMMENTS

Reviewer #1 (Remarks to the Author):

In their revised version, the authors have significantly improved the presentation of their data and have included a set of controls requested by the reviewers.

The inclusion of non-hydrolysable ATP in experiments measuring coiled-coil transitions has strengthened the claim that coiled-coil movements are driven by thermal fluctuations, rather than by ATP hydrolysis. The authors have also clarified that all dynamic movements strictly require the presence of DNA. Inclusion of DNA in the molecular dynamics simulations of NIPBL flexing has yielded a more modest energy difference, which can be reconciled with the overall energy balance of the system. These data make a strong case for the requirement for DNA to make head engagement energetically favourable, possibly preventing futile cycles of the complex. I also welcome that the current manuscript more accurately describes the results of the authors' experiments (e.g., translating head-hinge distances into bending angles) rather than presenting the reader already with interpretations. I furthermore appreciate that the manuscript better describes the available structural data.

However, I strongly disagree with the authors on the definition of a power stroke in the context of a molecular machine (see major comments 1 and 2). To avoid misinterpretation, the authors first need to carefully formulate their definition of a power stroke, which in the current context appears to apply to any chemically controlled conformational change, before using the term in the rest of the manuscript. It is also essential that the authors avoid the term "power stroke" in the abstract, since this promises data that the manuscript doesn't deliver (see major comment 3). Once this central concern has been addressed, I support publication of the revised manuscript in Nature Communications.

Major comments:

1. In response to reviewer 1, point 1 a-c:

I agree with the authors' conclusions that ATP-bound engaged heads can withstand forces up to 15 pN and that head dissociation presumably depends on ATP hydrolysis rather than on thermal fluctuations. However, these criteria don't fulfil the requirements for the definition of SMC head engagement as a "power stroke". For molecular motors, this term has been restricted to movements coupled to the substrate on which they perform work (e.g., actin filaments, microtubules, DNA, or even small molecule cargoes in the case of ABC transporters). The authors demonstrate neither that SMC head engagement performs work, nor that this conformational change is coupled to DNA translocation.

As mentioned before, to demonstrate the ability of a transition to perform work, the authors would need to show processive movement against an imposed force. The argument raised in the rebuttal letter that not all myosins or kinesins are processive ignores the key point that one must first prove that the conformational transition can perform work (non-zero directional displacement over time), before calling that transition a power stroke. In fact, both publications mentioned in the authors' response show that even non-processive kinesin and myosin directionally displace microtubule or actin filaments, respectively, against force and thus do perform work. While I agree that head engagement could – hypothetically – perform work in consecutive steps (as outlined in figure 1 of the authors' reply), the manuscript contains no data to demonstrate that it does.

2. In response to reviewer 1, point 6:

The authors aptly show that cohesin SMC heads under tension can come together in a 10-nm step. While it cannot be excluded that NIPBL acts as a mechanical platform to move the heads during engagement, the oscillation of the beads in this assay are sufficient to bridge this distance 5-10% of the time (trap stiffness 0.2 pN/nm). The data are thus fully compatible with a scenario where heads come together due to the thermal motion of the bead, dimerise by binding ATP and then withstand pulling forces until hydrolysis. Changing the force to 15 pN does not change the spacing of the heads or the bead oscillations (although resolution improves) and consequently does not affect the association rate. Above 15 pN, heads are pulled apart 30 nm and bead oscillations no longer bridge this distance (no associations detected).

Thus, the results cannot distinguish whether mechanical coupling is required or not during the engagement, i.e., whether ATP causes the heads to come together or whether the heads happen to come together and then bind ATP. This is at the heart of a true power stroke mechanism, where a “chemical” change (ATP binding) triggers a subsequent conformational change, instead of thermal motion allowing a conformation to be locked by ATP. Again, I would agree with the interpretation of the authors if they had measured consecutive directional steps and the effect thereupon by an external force, which is also the type of experimental setup that they refer to in Howard, 2006.

3. The presented data for ATP-driven head dimerisation cannot be equated to the power stroke movements in myosin and kinesin on their tracks, and the term should be carefully defined before it is used. An alternative reading in the abstract could be “the ATPase head engagement occurs in a single step of ~10 nm and is chemically driven, resisting forces up to 15 pN” or similar.

Minor comments:

1. Line 195: "interpolated" should read "extrapolated"
2. Line 197: change "shows" into "suggests"
3. Lines 370-373: consider removing this sentence.
4. Line 447: change "indicating that" into "compatible with..."
5. Line 457: "power stoke" should read "power stroke"
6. Line 460: change "AMPPNP does not support the head engagement" to "AMPPNP does not support dynamic head engagement"
7. Line 464: change "triggers" to "coincides with" or similar
8. Lines 471-476: The oscillations of the bead ($|x| = 4.5$ nm for trap stiffness of 0.2 pN/nm) are sufficient to allow heads to approach each other $\sim 10\%$ of the time under the experimental conditions and a required "strong mechanical connection" cannot be inferred. Remove these sentences.
9. Lines 478-484: This reasoning opposes the data, where NIPBL bending increases energy, rather than creating a new energy minimum. Consider removal or rephrasing.
10. Line 490: change "the ATPase head engagement" to "the transition from inward- to outward-facing conformation"
11. Lines 501-504: Redundant

12. Lines 520-568: Introduction of new data in the Discussion section and rather speculative, distracting from the main experimental message of the manuscript. Consider shortening and/or moving to supplementary figure legend.

13. Line 553: "then" should read "than"

14. Line 563: "coild" should read "coiled"

15. Line 578: change "implications on" to "implications for"

16. Figure 4 title: change "explain" to "proposed for"

17. The manuscript would benefit from some language editing.

Reviewer #2 (Remarks to the Author):

The authors have addressed all of my comments and I have no further concerns. The manuscript is ready for publication.

Reviewer #3 (Remarks to the Author):

All my remarks were sufficiently answered and I fully recommend publication of the article.

There is only one minor point that I noticed and that is that the labelling of Fig S1b seems wrong.

Reviewer #4 (Remarks to the Author):

The revised manuscript fully addresses all my concerns. It's a very solid study that provides a wealth of new data on cohesin dynamics. The new information about NIPBL and ATP (and ATP analog) dependence of the head-hinge dynamics strengthens the story. Force generation in head-head interactions and bending of NIPBL is much better presented and easier to follow. New/undated panels of Fig 1 and Fig 2 are very informative and helpful.

My only suggestion at this point is to simplify the text of the abstract. With 15nm/17nm/10nm numbers is sounds very technical. Also it's unclear from the abstract what are these distances (only later the read will find out that 15/17 are head-hinge and 10 is head-head).

REVIEWER COMMENTS

Reviewer #1 (Remarks to the Author):

In their revised version, the authors have significantly improved the presentation of their data and have included a set of controls requested by the reviewers.

The inclusion of non-hydrolysable ATP in experiments measuring coiled-coil transitions has strengthened the claim that coiled-coil movements are driven by thermal fluctuations, rather than by ATP hydrolysis. The authors have also clarified that all dynamic movements strictly require the presence of DNA. Inclusion of DNA in the molecular dynamics simulations of NIPBL flexing has yielded a more modest energy difference, which can be reconciled with the overall energy balance of the system. These data make a strong case for the requirement for DNA to make head engagement energetically favourable, possibly preventing futile cycles of the complex. I also welcome that the current manuscript more accurately describes the results of the authors' experiments (e.g., translating head-hinge distances into bending angles) rather than presenting the reader already with interpretations. I furthermore appreciate that the manuscript better describes the available structural data.

We thank the reviewer for the positive and enthusiastic assessment of our revised manuscript.

However, I strongly disagree with the authors on the definition of a power stroke in the context of a molecular machine (see major comments 1 and 2). To avoid misinterpretation, the authors first need to carefully formulate their definition of a power stroke, which in the current context appears to apply to any chemically controlled conformational change, before using the term in the rest of the manuscript. It is also essential that the authors avoid the term "power stroke" in the abstract, since this promises data that the manuscript doesn't deliver (see major comment 3). Once this central concern has been addressed, I support publication of the revised manuscript in Nature Communications.

We agree with reviewer that the term "power stroke" could be interpreted differently and therefore needs to be better defined. We followed the reviewer's suggestion, removed "power stroke" from the abstract and figure title. We also discussed limitations of the term and how it can be applied to cohesin given how it is used in the context of other motors. We thank the reviewer for their support in publication given the changes we have made.

Major comments:

1. In response to reviewer 1, point 1 a-c:

I agree with the authors' conclusions that ATP-bound engaged heads can withstand

forces up to 15 pN and that head dissociation presumably depends on ATP hydrolysis rather than on thermal fluctuations. However, these criteria don't fulfil the requirements for the definition of SMC head engagement as a "power stroke". For molecular motors, this term has been restricted to movements coupled to the substrate on which they perform work (e.g., actin filaments, microtubules, DNA, or even small molecule cargoes in the case of ABC transporters). The authors demonstrate neither that SMC head engagement performs work, nor that this conformational change is coupled to DNA translocation.

As mentioned before, to demonstrate the ability of a transition to perform work, the authors would need to show processive movement against an imposed force. The argument raised in the rebuttal letter that not all myosins or kinesins are processive ignores the key point that one must first prove that the conformational transition can perform work (non-zero directional displacement over time), before calling that transition a power stroke. In fact, both publications mentioned in the authors' response show that even non-processive kinesin and myosin directionally displace microtubule or actin filaments, respectively, against force and thus do perform work. While I agree that head engagement could – hypothetically – perform work in consecutive steps (as outlined in figure 1 of the authors' reply), the manuscript contains no data to demonstrate that it does.

We agree with reviewer that for molecular motors even non-processive ones, directional movement of the substrate has been shown, while we do not show the corresponding movement of DNA in our experiments.

We are delighted that the reviewer now finds our data convincingly showing that the head movement can generate force and that generation depends on ATP. We also agree that in the context of molecular motors, power strokes even in non-processive enzymes have been shown to unidirectionally displace substrate. This has not yet been shown for cohesin and went beyond the scope of this work. We have revised our text to make this clear.

2. In response to reviewer 1, point 6:

The authors aptly show that cohesin SMC heads under tension can come together in a 10-nm step. While it cannot be excluded that NIPBL acts as a mechanical platform to move the heads during engagement, the oscillation of the beads in this assay are sufficient to bridge this distance 5-10% of the time (trap stiffness 0.2 pN/nm). The data are thus fully compatible with a scenario where heads come together due to the thermal motion of the bead, dimerise by binding ATP and then withstand pulling forces until hydrolysis. Changing the force to 15 pN does not change the spacing of the heads or the bead oscillations (although resolution improves) and consequently does not affect the association rate. Above 15 pN, heads are pulled apart 30 nm and bead oscillations no longer bridge this distance (no associations detected).

Thus, the results cannot distinguish whether mechanical coupling is required or not during the engagement, i.e., whether ATP causes the heads to come together or whether the heads happen to come together and then bind ATP. This is at the heart of a true power stroke mechanism, where a “chemical” change (ATP binding) triggers a subsequent conformational change, instead of thermal motion allowing a conformation to be locked by ATP. Again, I would agree with the interpretation of the authors if they had measured consecutive directional steps and the effect thereupon by an external force, which is also the type of experimental setup that they refer to in Howard, 2006.

We thank the reviewer for acknowledging our finding that “SMC heads under tension can come together in a 10-nm step”. We agree that Brownian movement of the bead can potentially displace it occasionally by as much as 10 nm. However, our calculation shows that the probability of such an event must be significantly smaller than the estimate given by the reviewer. The time spent at distances of 10 nm or over should be proportional to

the normalized Boltzmann probability: $\frac{\int_{10}^{\infty} e^{-\frac{U}{k_B T}} dx}{\int_0^{\infty} e^{-\frac{U}{k_B T}} dx}$, where $U = \frac{1}{2} kx^2$. Therefore, for the trap

stiffness $k = 0.2$ pN/nm the probability of a bead fluctuating 10 nm or more should be ~ 0.002 . This is consistent with our data, which shows that excursions by as much as 10 nm almost never happen even for smaller trap stiffnesses (Supplementary Fig. S1b top right graph).

More importantly, however, reviewer’s suggestion that thermal fluctuations could bridge the 10 nm distance for cohesin head domains at all forces is inconsistent with our data presented on Fig. 2. This is because the association (engagement) rate depends not only on the distance (as the reviewer correctly pointed out) but also equally on force via Arrhenius factor: $\exp(F \cdot \Delta / k_B T)$, where Δ is 10 nm, F is applied force and $k_B T = 4.14$ pN·nm Boltzmann factor. If thermal fluctuations could bridge 10 nm distance against 10 pN force, then the likelihood of bridging the same 10 nm distance at 15 pN force would decrease by a factor of $\exp(10 \cdot 15 / 4.14) / \exp(10 \cdot 10 / 4.14) \approx 1.8e5$. This dramatic difference in rates would be expected every time the force changes by 5 pN. This is inconsistent with our experiments, which show that the association rate does not depend on force (Figure 2f,g). This demonstrates that thermal fluctuations alone cannot explain the head engagement under external force, and this was the original reason of our suggestion that ATPase head engagement may generate a “power stroke”.

We, nonetheless, agree that our experiments do not show consecutive directional steps, which have been shown for other motors generating power stroke. We discussed this difference in the main text and removed “power stroke” where the difference was not clear.

3. The presented data for ATP-driven head dimerisation cannot be equated to the power stroke movements in myosin and kinesin on their tracks, and the term should be carefully defined before it is used. An alternative reading in the abstract could be “the

ATPase head engagement occurs in a single step of ~ 10 nm and is chemically driven, resisting forces up to 15 pN" or similar.

We thank the reviewer for this suggestion, which we followed. We have revised the abstract based on the reviewer's suggestion, defined what we meant by power stroke and discussed the limitations of using this term in the light of mechanisms used by other motors like myosin and kinesin.

Minor comments:

1. Line 195: "interpolated" should read "extrapolated"
2. Line 197: change "shows" into "suggests"
3. Lines 370-373: consider removing this sentence.
4. Line 447: change "indicating that" into "compatible with..."
5. Line 457: "power stoke" should read "power stroke"
6. Line 460: change "AMPPNP does not support the head engagement" to "AMPPNP does not support dynamic head engagement"
7. Line 464: change "triggers" to "coincides with" or similar

We thank reviewer for the comments 1-7, which we all accepted and changed the text accordingly in the revised manuscript.

8. Lines 471-476: The oscillations of the bead ($|x| = 4.5$ nm for trap stiffness of 0.2 pN/nm) are sufficient to allow heads to approach each other $\sim 10\%$ of the time under the experimental conditions and a required "strong mechanical connection" cannot be inferred. Remove these sentences.

We agree with reviewer that "strong mechanical connection" cannot be inferred from our data, which we made clear in the revised text. We removed part of the sentences and left two, which we clearly indicated as speculation.

9. Lines 478-484: This reasoning opposes the data, where NIPBL bending increases energy, rather than creating a new energy minimum. Consider removal or rephrasing.

We thank the reviewer for spotting this logical inconsistency. We have rephrased the paragraph to make it clearer.

10. Line 490: change "the ATPase head engagement" to "the transition from inward- to outward-facing conformation"

We thank reviewer for the suggestion. We changed "the ATPase head engagement" to the "conformational change".

11. Lines 501-504: Redundant

We removed the redundant text.

12. Lines 520-568: Introduction of new data in the Discussion section and rather speculative, distracting from the main experimental message of the manuscript. Consider shortening and/or moving to supplementary figure legend.

We thank the reviewer for this suggestion. We have significantly shortened this part and moved distracting material to the supplementary figure legends and methods.

13. Line 553: "then" should read "than"
14. Line 563: "coild" should read "coiled"
15. Line 578: change "implications on" to "implications for"
16. Figure 4 title: change "explain" to "proposed for"
17. The manuscript would benefit from some language editing.

We thank the reviewer for the comments 13-17, which we have all followed and revised the manuscript accordingly.

Reviewer #2 (Remarks to the Author):

The authors have addressed all of my comments and I have no further concerns. The manuscript is ready for publication.

We thank the reviewer for the positive assessment of our work!

Reviewer #3 (Remarks to the Author):

All my remarks were sufficiently answered and I fully recommend publication of the article.

We thank the reviewer for the positive assessment of our work!

There is only one minor point that I noticed and that is that the labelling of Fig S1b seems wrong.

We thank the reviewer for spotting this typo, which we corrected.

Reviewer #4 (Remarks to the Author):

The revised manuscript fully addresses all my concerns. It's a very solid study that provides a wealth of new data on cohesin dynamics. The new information about NIPBL and ATP (and ATP analog) dependence of the head-hinge dynamics strengthens the story. Force generation in head-head interactions and bending of NIPBL is much better

presented and easier to follow. New/undated panels of Fig 1 and Fig 2 are very informative and helpful.

My only suggestion at this point is to simplify the text of the abstract. With 15nm/17nm/10nm numbers is sounds very technical. Also it's unclear from the abstract what are these distances (only later the read will find out that 15/17 are head-hinge and 10 is head-head).

We thank the reviewer for the positive assessment of our work and their suggestion! We have simplified the abstract to make it clearer.